# Spatial transcriptomics of a parasitic flatworm provides a molecular map of drug targets and drug resistance genes

Svenja Gramberg ®[1], Oliver Puckelwaldt ®[1], Tobias Schmitt[1], Zhigang Lu[2] & Simone Haeberlein ®[1] ✉

The spatial organization of gene expression dictates tissue functions in multicellular parasites. Here, we present the spatial transcriptome of a parasitic flatworm, the common liver fluke *Fasciola hepatica*. We identify gene expression profiles and marker genes for eight distinct tissues and validate the latter by in situ hybridization. To demonstrate the power of our spatial atlas, we focus on genes with substantial medical importance, including vaccine candidates (Ly6 proteins) and drug resistance genes (glutathione S-transferases, ABC transporters). Several of these genes exhibit unique expression patterns, indicating tissue-specific biological functions. Notably, the prioritization of tegumental protein kinases identifies a PKCβ, for which small-molecule targeting causes parasite death. Our comprehensive gene expression map provides unprecedented molecular insights into the organ systems of this complex parasitic organism, serving as a valuable tool for both basic and applied research.

*Fasciola hepatica*, together with related species, is the causative agent of fascioliasis, a zoonotic disease and food-borne infection that compromises liver function and affects at least 2.4 million people and numerous livestock worldwide[1]. As typical for parasitic flukes (trematodes), *Fasciola spp.* have a complex life cycle, which includes an intermediate snail host and a mammalian definite host[2]. The definitive host becomes infected by ingesting encysted larvae, so-called metacercariae. In the intestine, the newly excysted juveniles (NEJs) hatch from the cysts, penetrate the intestinal wall and migrate through the liver tissue until they have developed into mature adult flukes that reach remarkable sizes of up to 3 cm in length[2]. Adult parasites live in the bile ducts of the host's liver for several years, where they produce an impressive number of up to 50,000 eggs per day[3]. Adult *F. hepatica* are dorso-ventrally flattened, leaf-like in shape and composed of a skin-like tegument, two muscular suckers, a branched intestine, complex reproductive organs and further, largely uncharacterized tissues[2].

The introduction of "omics" technologies into parasite research has accelerated the study of key molecules involved in the biology, pathogenicity and virulence of liver flukes[4]. However, there is a lack of information on tissue-specific gene expression within the parasites, and many aspects of liver fluke biology remain poorly understood: Which genes are essential for the survival of the worm? Which ones serve the parasite-host interaction? Which molecular processes control the reproduction of the worm and thus ensure the persistence of the parasite cycle? This lack of knowledge also complicates the development of new treatments for fascioliasis. To date, triclabendazole (TCBZ) is the only drug that is effective against almost all intra-mammalian life stages of liver flukes, but reports of TCBZ-resistant parasite strains are increasing[5]. This drives global research endeavors to find alternative treatments and effective vaccines[4,6].

Information on gene expression in individual parasite tissues or cells would allow us to predict the usefulness of selected proteins as drug or vaccine targets and thereby facilitate a more strategic drug- and vaccine-target search in *F. hepatica*. It is assumed that proteins expressed in the body surfaces of trematodes, tegument and intestine, are particularly suitable as drug and vaccine targets[7,8]. These organs are crucial for the maintenance of body homeostasis and thus ensure the survival of the worm. They also supply all other body cells with

[1]Institute of Parasitology, Justus Liebig University Giessen, Giessen, Germany. [2]Institute of Food Science and Biotechnology, University of Hohenheim, Stuttgart, Germany. ✉e-mail: Simone.Haeberlein@vetmed.uni-giessen.de

nutrients and are directly involved in the parasite-host interaction[9,10]. In addition, the body surfaces are particularly accessible for compounds, which should improve the effectiveness of medication.

Spatial transcriptomics technologies are capable of providing transcriptome-wide and spatially resolved gene expression data in tissue context[11]. This has demonstrated high value for enhancing the understanding of metazoan biology ranging from humans to insects and plants[12–14]. This method has also the power to improve our understanding of multicellular pathogens[15]. In order to gain deeper insights into liver fluke biology and to address some of the above questions, we created a spatial transcriptome of adult *F. hepatica*. We characterized and compared gene expression patterns of eight distinct parasite tissues and revealed tissue-preferential expression of vaccine candidate and drug resistance genes. Furthermore, we exemplified the usefulness of this new resource by prioritizing tegument- and gut-expressed genes for a drug repurposing approach. With this work, we provide a dataset that enables a rapid and uncomplicated evaluation of the spatial expression of thousands of liver fluke genes serving as a source of inspiration for both fundamental questions and the development of new therapeutic strategies.

## Results

### Identification of eight transcriptionally distinct tissues in adult liver flukes

Employing the 10x Genomics Visium technology, we constructed a transcriptomic map of the adult stage of *F. hepatica*, the life stage causing chronic liver disease. In order to achieve maximum release of high-quality RNA from cryosections of the parasite, we first optimized the tissue permeabilization time using the 10x Visium tissue optimization workflow (Supplementary Fig. 1a). Subsequently, we processed four transversal cryosections, each containing a different set of tissues (Supplementary Fig. 2a), using the 10x Visium spatial gene expression platform and Illumina sequencing to obtain spatially resolved gene expression data from those sections. In this manuscript, the term "expression" refers to transcript levels, not protein levels. An overview of the workflow is shown in Fig. 1a.

All sections together covered a total of 2020 mRNA-binding spots, each coated with millions of barcoded oligonucleotides. We captured a median of 2192 genes and 6138 UMIs (unique molecular identifiers) per spot (Supplementary Fig. 2g, h, Supplementary Data 1). In total, over all spots, we detected transcripts of 9847 different genes, constituting 79.3% of all gene transcripts in the *F. hepatica* genome (PRJNA179522)[16]. We then used Seurat[17,18] to perform clustering and to identify transcriptionally distinct tissues. In this way, we received individual clusters representing eight tissues: tegument (561 spots), gut (154 spots), parenchyma (410 spots), vitellarium (279 spots), uterus (134 spots), ovary (92 spots), testis (354 spots) and Mehlis' gland (36 spots) (Fig. 1b, c, Supplementary Fig. 2c–f). Due to the given resolution of the Visium approach (55 μm spot diameter, 100 μm spot-to-spot distance[19]), some cell types that are spatially close to each other could not be discriminated and were combined in one cluster (Fig. 1d, e). This applies, for example, to the tegument cluster, which includes all subtegumental cells, such as subtegumental muscle cells and possibly also subtegumental neurons in addition to tegumental cytons (Supplementary Fig. 3).

We next used Seurat to identify differentially expressed genes that were best suitable to characterize the different tissues in the dataset (further referred to as "marker genes"). Figure 2a shows that each tissue possessed a set of cluster-defining markers whose average expression in the relevant cluster was significantly above the mean expression of this gene in all other clusters (see Supplementary Data 2 for the full list of marker genes). To verify the robustness of selected markers, we used knowledge from previous publications, related organisms, and classical ISH experiments. Known tissue markers such as cathepsin L[20], leucine aminopeptidase[21], legumain[22] and saposin B[23]

for the intestine, calcium-binding proteins for the tegument[24,25], fatty-acid-binding proteins in the parenchyma[26] and vitelline protein B1 for vitellarium and eggs[27] were also present among the markers of these tissues in our dataset (Supplementary Data 2). For the most studied trematode, *Schistosoma mansoni*, spatial transcriptomics data are not available, but we were able to compare our data to existing scRNAseq data. Several marker genes highlighted in our study, for example, parenchymal cathepsin B, ovarian bmpg and vitelline tyrosinase 1 and 2 were also found among the marker genes for the same tissues in *S. mansoni*[28,29]. These tissue-specific gene signatures therefore appear to be evolutionarily conserved among parasitic flatworms. In addition, we performed in situ hybridizations for 21 genes, reconfirming the expression patterns of known markers and also identifying previously unknown markers for different liver fluke tissues. These will be highlighted in the following results sections.

Gene Ontology (GO) enrichment analysis of marker genes finally revealed characteristic biological processes and molecular functions for each cluster (Fig. 2b, c). For instance, the gut cluster exhibited a significant enrichment in genes associated with the term "proteolysis." Testis and ovary clusters were enriched in genes involved in "microtubule-based movement" and various biosynthetic processes, respectively. These analyses suggested that each cluster is molecularly distinct and that our dataset is capable of displaying the different biological functions of different tissue types.

### Spatial co-expression analysis reveals common features of liver fluke gonads

The marker analysis in Seurat aims to identify genes that are characteristic of a specific tissue cluster. However, we were also interested in identifying genes with characteristic spatial expression patterns beyond the boundaries of individual clusters. Therefore, we performed a spatial co-expression analysis with Giotto[30]. Giotto represents spatial relationships between different spots as a spatial network (Fig. 3a). Within this network, Giotto identified the 2500 most spatially coherent genes, which were then selected to create a co-expression matrix. Clustering resulted in 15 co-expression modules whose spatial expression patterns were summarized and visualized as metagenes (Fig. 3b, d, Supplementary Fig. 4, Supplementary Data 4). These spatial metagene profiles turned out to be similar to the known anatomical structures of the liver fluke and therefore largely corresponded to our tissue clusters from the Seurat analysis (Fig. 3b, c, Supplementary Fig. 4). Metagenes 1 & 5, however, were not limited to single organs, but combined genes that were expressed in both the testis and the ovary (Fig. 3b, c). Liver flukes are hermaphrodites, and male and female reproductive organs make up a large proportion of the adult fluke[31], mirroring their exceedingly high fecundity, which allows the parasites to spread efficiently among hosts[3]. GO term enrichment analysis showed that several of these spatially correlated genes detected in both gonads are involved in mitosis, cell cycle and DNA repair (Fig. 3e). Against our expectations, meiosis was not among the enriched GO terms. This is probably due to an annotation gap in available GO term annotation data. When we manually browsed the list of co-expressed genes, we identified several genes, e.g., encoding HORMA domain-containing protein 2 (HORMAD2, D915_003478) and a synaptonemal complex protein (SYCP2/D915_003691), which are thought to be involved in meiotic chromosome segregation[32] (Supplementary Data 4).

Based on the results from the Giotto analysis, we were interested in the expression of further stem cell- and cell cycle-associated genes in our dataset. A list of cell cycle-associated genes was taken over from Robb et al.[33] (Supplementary Data 5). Among others, this list encompassed components of the conserved MCM2-7 complex (D915_009918, D915_01033, D915_006290, D915_005936), cell division cycle genes (*cdc-20*/D915_006257, *cdc-45*/D915_007655), as well as histone s (D915_002864, D915_002825) and DNA polymerases (D915_004675,

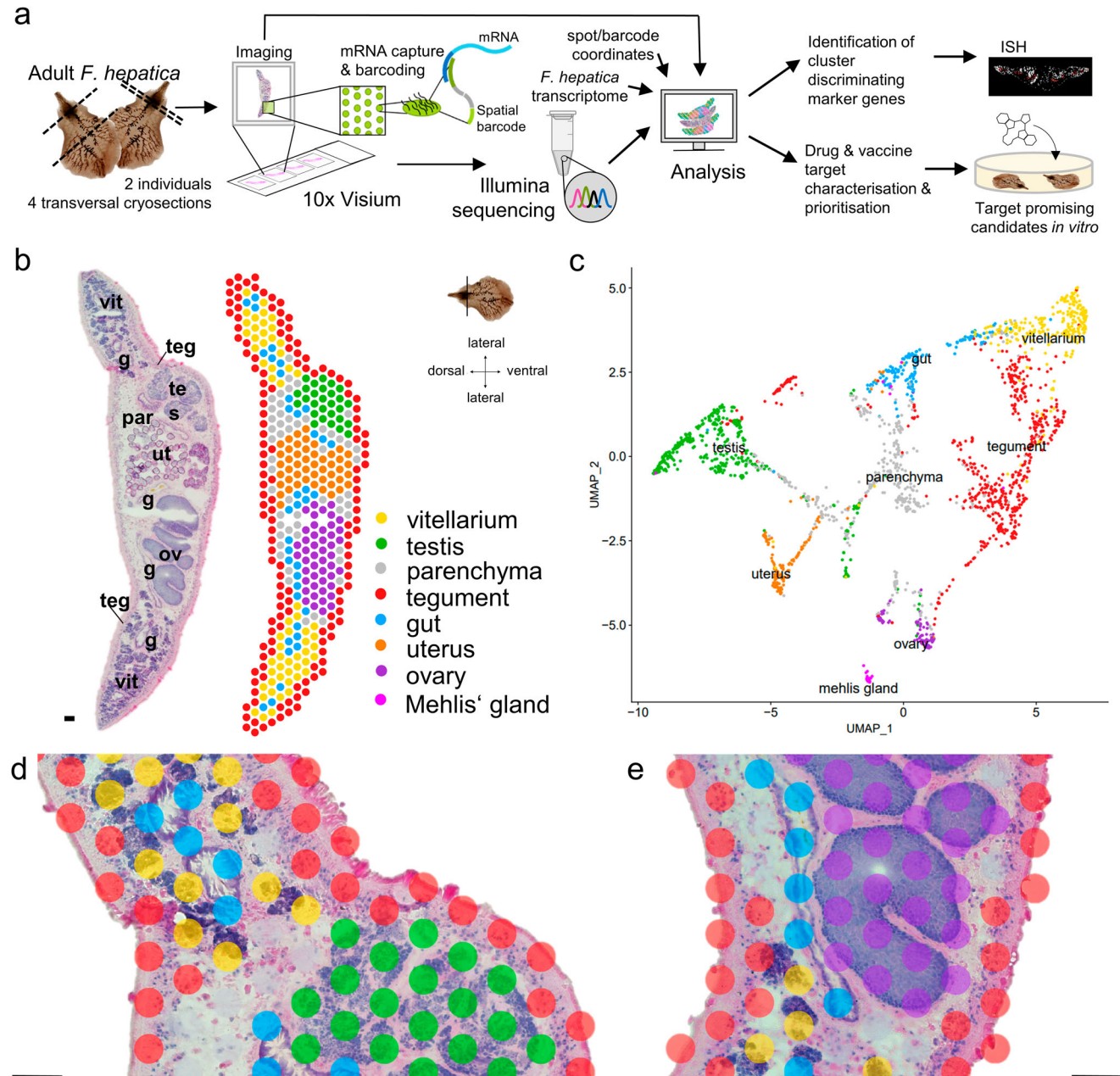

**Fig. 1 | A spatial transcriptome of *F. hepatica* cross sections. a** Scheme describing the experimental workflow: Four transversal cryosections of two adult liver flukes were placed on a 10x Visium Spatial Gene Expression slide, stained and imaged. mRNA release, barcoding and sequencing were performed according to the 10x Visium protocol. During the analysis, all transcripts were mapped back to their corresponding spots on the slide and annotated using the reference transcriptome. Clustering was carried out to identify transcriptionally distinct tissues and tissue-specific markers. Selected markers were validated by in situ hybridization (ISH). The dataset was then used to explore spatial expression profiles of drug and vaccine candidates. One promising candidate was finally targeted with a small-molecule compound in vitro. **b** H&E-stained transversal tissue section and corresponding spatial projection of 412 mRNA-binding spots covered by this tissue section. Clusters are colored and labeled. Sectioning plane and orientation are indicated on the right. The image is representative of the four tissue sections used in the workflow. Please note: There is no Mehlis' gland in this tissue section. See Supplementary Fig. 2 for H&E stainings and spatial projections of the remaining three tissue sections. g: gut, ov: ovary, par: parenchyma, teg: tegument, tes: testis, ut: uterus, vit: vitellarium. Scale = 100 μm. **c** Uniform Manifold Approximation and Projection (UMAP) of 2020 spots derived from four different *F. hepatica* cross sections. Clusters are colored and labeled according to (**b**). **d, e** Magnified view of an overlay of the H&E-stained tissue section and the spatial cluster projection shown in (**b**). mRNA capturing spots have a diameter of 55 μm and therefore span multiple cells and sometimes different tissues. As an example, see Supplementary Fig. 3 for expression patterns of known tegument and muscle markers. Clusters are colored and labeled according to (**b**). Scale = 100 μm.

D915_001192, D915_003363). Although it is to be expected that cell proliferation also occurs elsewhere in the worm's tissue, e.g., for renewal of the tegument or intestinal epithelium, the stem cell- and cell cycle-associated genes examined here were predominantly expressed in the gonads of the fluke (Fig. 3f). Of the 79 genes in the list, 55 genes were expressed above average in the testis and even 74 in the ovary. One exception was *p53-1* (D915_001973), for which expression was found to be enriched in the tegument cluster (Fig. 3f). This matches the

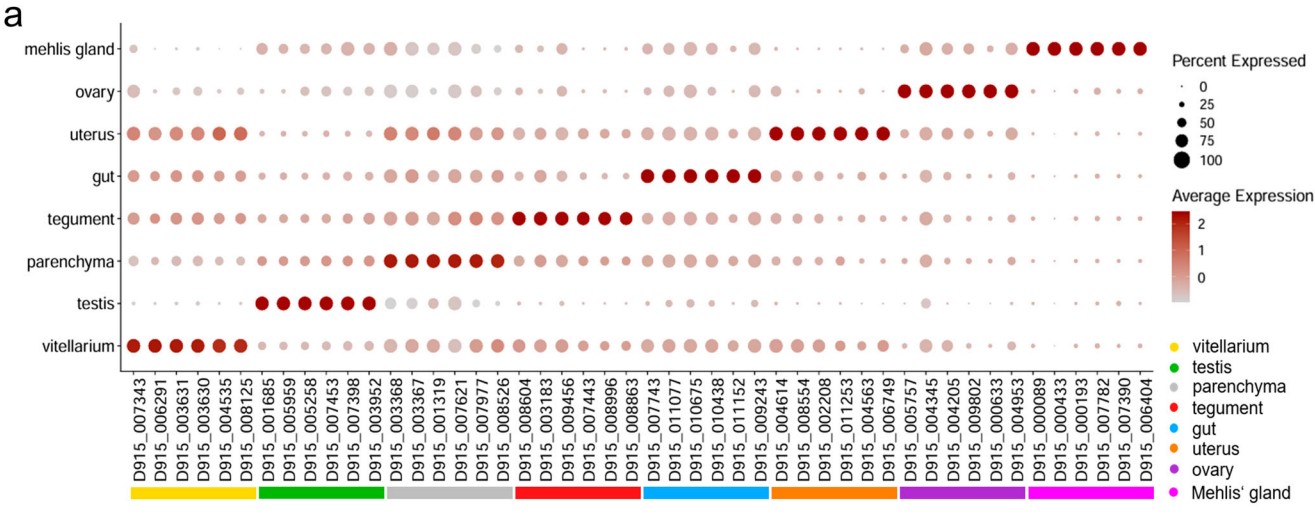

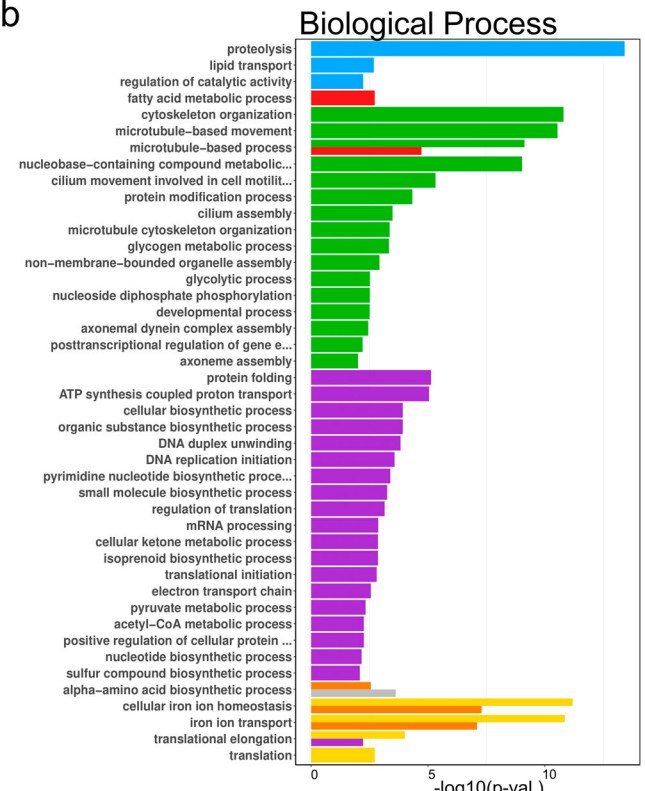

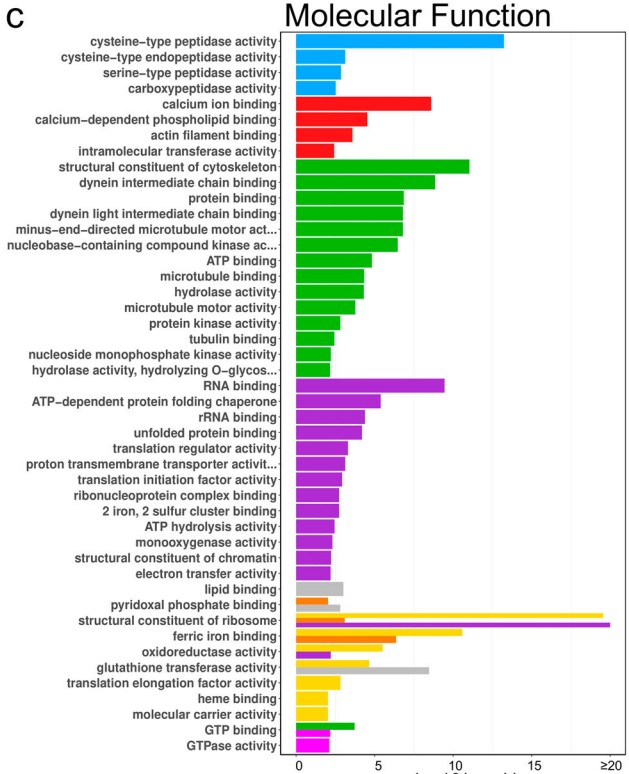

**Fig. 2 | Eight transcriptionally distinct tissues in adult liver fluke cross sections.** **a** DotPlot showing expression profiles of the top 6 marker genes (sorted by "predictive power") for all eight tissues in the spatial transcriptomics dataset. Dot color encodes the average expression level (mean of UMI counts, normalized and scaled) across all spots within a cluster. Dot size encodes the percentage of spots within a cluster that have captured this transcript. **b**, **c** Gene ontology analysis of marker genes (top 75% per cluster) revealed characteristic biological processes (**b**) and molecular functions (**c**) for each cluster. Overrepresented functional terms for each cluster were identified using a two-sided Fisher's exact test ($p$-value < 0.05). Bars for individual clusters are colored according to legend and labeling in (**a**). FOR STRING analyses of marker genes of the ovary and testis clusters, see Supplementary Fig. 7. **a**–**c** Source data are provided as a Source Data file.

findings of Wendt et al., who demonstrated that the schistosome *p53-1* orthologue is an important regulator of tegument differentiation[34]. The enrichment of stem-cell and cell-cycle markers in gonads is consistent with the fact that the testis and ovary cluster showed the highest transcriptional activity and the largest number of expressed genes compared to all other tissues (Supplementary Fig. 2c–h). Both together demonstrate a high metabolic activity and rapid rate of cellular differentiation and turnover in germ cell-forming organs of liver flukes.

**Seurat identifies markers of liver fluke reproductive tissues**

Our analysis provided further insights into the gene expression of the liver fluke reproductive system. Using Seurat, we were able to transcriptionally characterize the egg-production apparatus, including vitellarium, uterus and Mehlis' gland (Supplementary Figs. 5 and 6), and to describe markers for the gonads of liver flukes (Fig. 4).

GO-term and STRING analysis for the ovary cluster showed that detected transcripts correspond to proteins involved in a variety of

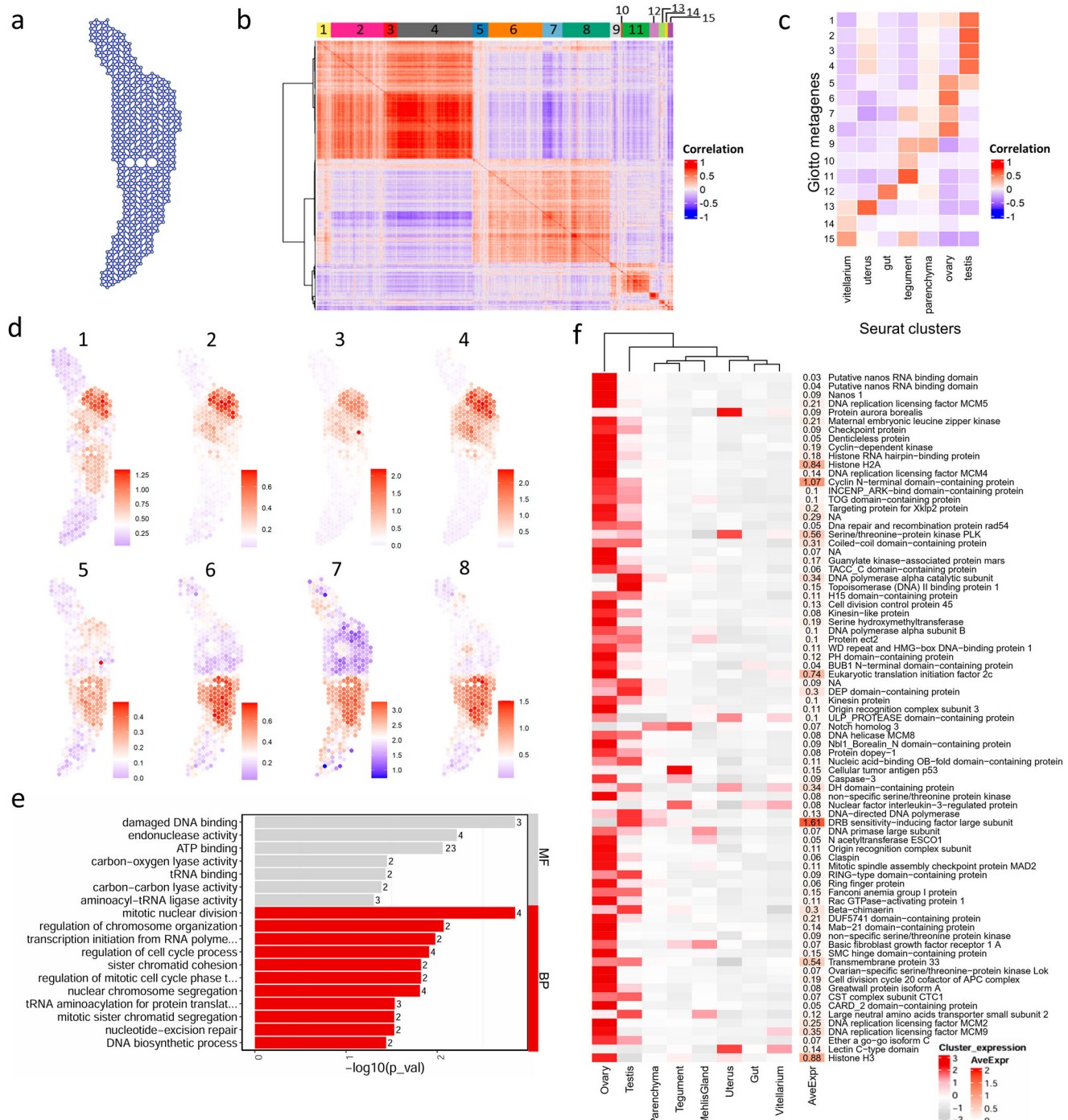

**Fig. 3 | Spatial co-expression analysis revealed shared expression of stem cell- and cell cycle-associated genes in gonadal tissues. a** Giotto represents spatial relationships between different spots as a spatial network, which is required to identify individual genes with spatial coherent expression patterns. **b** Heatmap showing the correlation of spatial expression patterns among the 2500 most spatially coherent expressed genes in the dataset. Groups of genes with similar expression profiles were clustered into 15 spatial co-expression modules, which are indicated with different colors on top. **c** Heatmap showing correlation of Seurat cluster assignment and Giotto co-expression modules. **d** Metagene visualizations for all spatial co-expression modules associated with gonads in (**c**). See Supplementary Fig. 4 for visualizations of metagenes 9–15. **e** Gene ontology enrichment analysis of genes contained in metagenes 1 & 5 revealed characteristic biological processes and molecular functions. Overrepresented functional terms were identified using a two-sided Fisher's exact test (*p*-value < 0.05). **f** Left: Heatmap of 79 stem cell- and cell cycle-associated genes showing their average expression per cluster (mean of UMI counts, normalized and scaled). Right: Heatmap showing the average spot expression (AveExpr) of those genes in the whole dataset (log1p normalized counts). **b**, **c**, **e**, **f** Source data are provided as a Source Data file.

biosynthetic processes, in particular (ribo-) nucleotide and small-molecule synthesis, DNA replication and translation (Fig. 2b and Supplementary Fig. 7c). In addition, we identified two C-type lectins of unknown function (D915_005862, D915_005757) with distinct spatio-temporal expression during oocyte differentiation (Fig. 4f, i). The liver

fluke ovary is structured in a way that oogonia and early primary oocytes reside in the periphery of the ovarian tubule while late primary oocytes are found in the center[35]. By ISH, we showed that the bone marrow proteoglycan (BMPG, D915_005862) was predominantly expressed in early primary oocytes, but far less in late primary oocytes

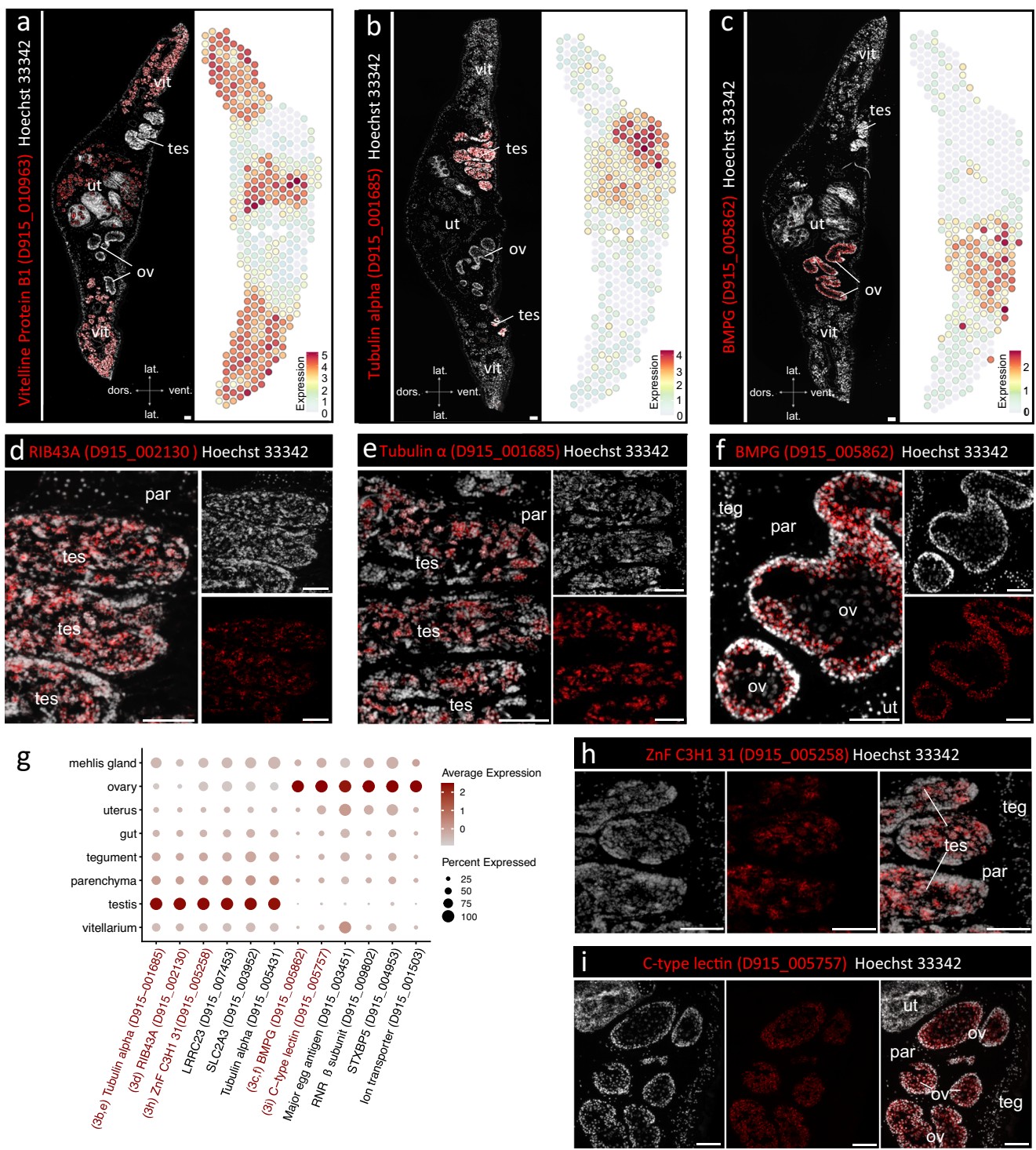

**Fig. 4 | Spatial expression of marker genes for liver fluke reproductive organs.**
**a–c** Fluorescent in situ hybridization (FISH) overview (left) and spatial projection (right) of the vitelline protein B1 (D915_010963) (**a**), a testis-specific tubulin alpha (D915_001685) (**b**) and the ovary marker bone marrow proteoglycan (BMPG, D915_005862) (**c**). Section orientation is indicated at the bottom (dors: dorsal, vent: ventral, lat: lateral). Expression level encoded by color (gray = low, red = high). See Supplementary Figs. 5 and 6 for more details and markers of the egg-production apparatus (vitellarium, uterus and Mehlis' gland). See Supplementary Fig. 8 for tissue expression patterns of liver fluke tubulins. **d** FISH of the ribbon protein RIB43A (D915_002130). **e, f** Magnified view of FISH stainings for genes shown in (**b**) and (**c**), respectively (different tissue section, same experiment). **g** DotPlot showing expression profiles of selected marker genes for testis and

ovary. Dot color encodes the average expression level (mean of UMI counts, normalized and scaled) across all spots within a cluster. Dot size encodes the percentage of spots within a cluster that have captured this transcript. Please note: While spatial plots (**a–c**) are shown for only one representative section, the DotPlot includes expression data from all four tissue sections in the dataset. Genes labeled in red were validated by FISH. Figure panels showing the respective FISH experiment are indicated. Source data are provided as a Source Data file. **h** FISH of Zinc finger C3H1 domain-containing protein 31 (D915_005258). **i** FISH of C-type lectin domain-containing protein (D915_005757). **a–f, h, i** Scale = 100 μm. ov: ovary, par: parenchyma, teg: tegument, tes: testis, ut: uterus, vit: vitellarium. For numbers of ISH experiments performed for each gene, see "Statistics and reproducibility" and Supplementary Data 9.

(Fig. 4f), while D915_005757 transcripts were also detected in late primary oocytes (Fig. 4i).

The production machinery of the testis is clearly directed toward one goal: the production of large numbers of motile spermatozoa. STRING analysis for marker genes of the testis cluster displayed a tight network of protein-protein associations (Supplementary Fig. 7a). The network was functionally enriched in numerous terms associated with "Cytoskeleton", "Microtubule", "Cilium" and "Axoneme". Markers included several alpha and beta tubulins and the *rib43a* gene encoding a flagellar ribbon protein (D915_002130) (Fig. 4g, Supplementary Data 2). ISH demonstrated that the alpha tubulin D915_001685 and *rib43a* were expressed in almost all stages of spermatogenesis, apart from spermatogonia, which are located in the periphery of the testicular tubules[33] (Fig. 4b, d, e). Tubulins are of continuous research interest as they are molecular targets of triclabendazole, a benzimidazole and the drug of choice to treat fasciolosis[5]. A closer look at the expression patterns of known alpha- and beta-tubulin genes in combination with published life-stage expression data revealed correlations between testicular expression and upregulation in maturing parasites (see Supplementary Fig. 8 for details).

To ensure successful germ cell formation, cellular processes must be tightly regulated. Against this background, we noticed an enriched expression of seven and eight different Zinc finger proteins in the ovary and testis, respectively (Supplementary Data 2). These included several C2H2 class proteins and CCCH domain-containing proteins (D915_005258, D915_003685), which stand out among zinc fingers as they bind RNA, not DNA, and thereby regulate RNA metabolism[36]. Indeed, the CCCH zinc finger D915_003685 was part of a STRING subnetwork of ovary marker genes functionally enriched in genes associated with "mRNA processing" and "RNA splicing" (Supplementary Fig. 7d). Additionally, ISH was used to confirm the expression of Zinc finger CCCH domain-containing protein 31 (D915_005258) in the testis (Fig. 4h).

Taken together, the combined use of Giotto and Seurat allowed us to characterize both common features and individual characteristics of the two gonads of liver flukes.

## Characterizing somatic tissues of liver flukes using spatial transcriptomics

Trematodes have developed several functional and morphological adaptations to a parasitic lifestyle. Markers for the three somatic tissues in our dataset, gut, tegument and parenchyma, allowed us to infer their diverse biological functions. For instance, the fluke's gut is well equipped to digest host erythrocytes and hemoglobin and thereby provide amino acids needed for the production of 25,000 eggs per day[19]. This digestive function was well reflected by 17 digestive enzymes, mainly proteases and hydrolases, among the top 50 marker genes (Supplementary Data 2). Expression of the two cysteine proteases legumain (D915_002224) and cathepsin L (D915_011077) could be specifically allocated to the intestinal epithelium via ISH (Fig. 5a, b, g).

The syncytial tegument is another remarkable feature of parasitic flatworms. Similar to the gut, it serves absorptive functions, but it also acts as a protective layer at the host-parasite interface[10]. Related to nutrient import, the expression of a glucose transporter (D915_005316) and an amino acid transporter (D915_001928) was enriched within the tegument cluster (Supplementary Data 2). The stability of the parasite's outer layer is ensured by multiple cytoskeletal and membrane-associated proteins such as a cytoplasmic type actin (D915_007443) and a tetraspanin family protein (D915_000797) (Fig. 5i, Supplementary Data 2). ISH detected their transcripts within groups of cells sitting below the body wall musculature with cytoplasmic protrusions toward the syncytial layer (Fig. 5h, j). Analysis of annotated GO terms for the tegument cluster further showed enrichment of molecular functions associated with "calcium ion binding" (Fig. 2c). This GO term is represented by three annexins, one calmodulin 3, one alpha-

actinin and seven EF hand domain-containing proteins (Supplementary Data 2). ISH for the EF hand domain-containing protein D915_003074 showed a similar expression pattern as for the structural proteins mentioned above (Fig. 5c, d). Tegumental EF hand domain-containing calcium-binding proteins are an unusual protein family unique to parasitic flatworms[37]. The exact function of these proteins is still unclear, but it is assumed that they play an important role in regulating the diverse cytoskeletal processes of the tegument[37].

The parenchyma is a specialized tissue in flatworms embedding all other organs. To obtain an overview of its suspected biological role, we performed GO term analysis, which suggested lipid and amino acid metabolism as two of the main functions (Fig. 2b, c). Related to these metabolic functions, several fatty acid-binding proteins (FABP, D915_003368, D915_003367, D915_008422) and three de- or transaminases (D915_004674, D915_004407, D915_008390) were among the markers of the parenchyma. Furthermore, we found two cysteine proteases, a cathepsin L (D915_005616) and a cathepsin B (D915_007096), which, in contrast to related enzymes, were not only found expressed in the intestine, but predominantly in the parasite's parenchyma (Fig. 5i and Supplementary Data 2). Cathepsin expression in parenchymal cells was previously described for schistosomes[28], but not for liver flukes, and we were able to confirm cathepsin B expression in parenchymal cells by ISH (Fig. 5e). Another marker of the liver fluke parenchyma is a heparan sulfate proteoglycan (HSPG, an extracellular matrix protein) (D915_000229), whose parenchymal expression was also confirmed by ISH (Fig. 5f). Liver fluke HSPG possesses similarities to human HSPG2, which is a functionally diverse protein whose different domains are able to bind other extracellular matrix components, cells, LDL and growth factors. Another prominent GO term of the parenchyma cluster was linked to "glutathione transferase activity". Glutathione S-transferases (GSTs) are among the molecules that defend the parasite against immune-induced damage and may also mediate cellular detoxification of drugs[38]. This important gene family is therefore addressed in more detail in a separate results section. A defense function of the parenchyma is further supported by the expression of helminth-defense molecule 1 (HDM1/MF6, D915_007621) (Fig. 5i, k). Our data thus supports the idea that the parenchyma acts both as a flexible skeleton and as a site for metabolism, storage, and transport of nutrients[9,26], but also revealed a prominent expression of defense-related proteins.

A major strength of spatial transcriptomics is that expression patterns of large gene families can be explored at once without further experimental effort. To demonstrate this capacity, we explored the spatial expression of two large gene families, GSTs and ABC transporters, which are both associated with defense, detoxification and drug resistance.

## Spatial distribution of GSTs supports a specialized role of the parenchyma in detoxification

The glutathione S-transferases (GSTs) represent an important group of enzymes that detoxify both endogenous compounds and foreign chemicals such as antiparasitic drugs[38,39]. For *F. hepatica*, GSTs out of four classes (Mu, Sigma, Omega and Zeta) have been identified and characterized by biochemistry and bioinformatics[40–42]. Our work complements these findings with information on the spatial expression of 11 cytosolic (6x Mu, 2x Sigma, 2x Omega, 1x Zeta) and two microsomal GSTs (Fig. 6a, c, Supplementary Fig. 9a). Phylogenetic analysis of their sequences together with human and known *Fasciola* GST sequences confirmed isoform assignment (Supplementary Fig. 10a).

GSTs have been reported to be widely distributed in liver fluke tissues, particularly in the parenchyma[43,44]. However, localization of individual isoforms, especially Mu-class GSTs, was hampered by high sequence similarities and cross-reactivity of antisera[45]. By using Visium, a sequencing-based technology, we were now able to distinguish

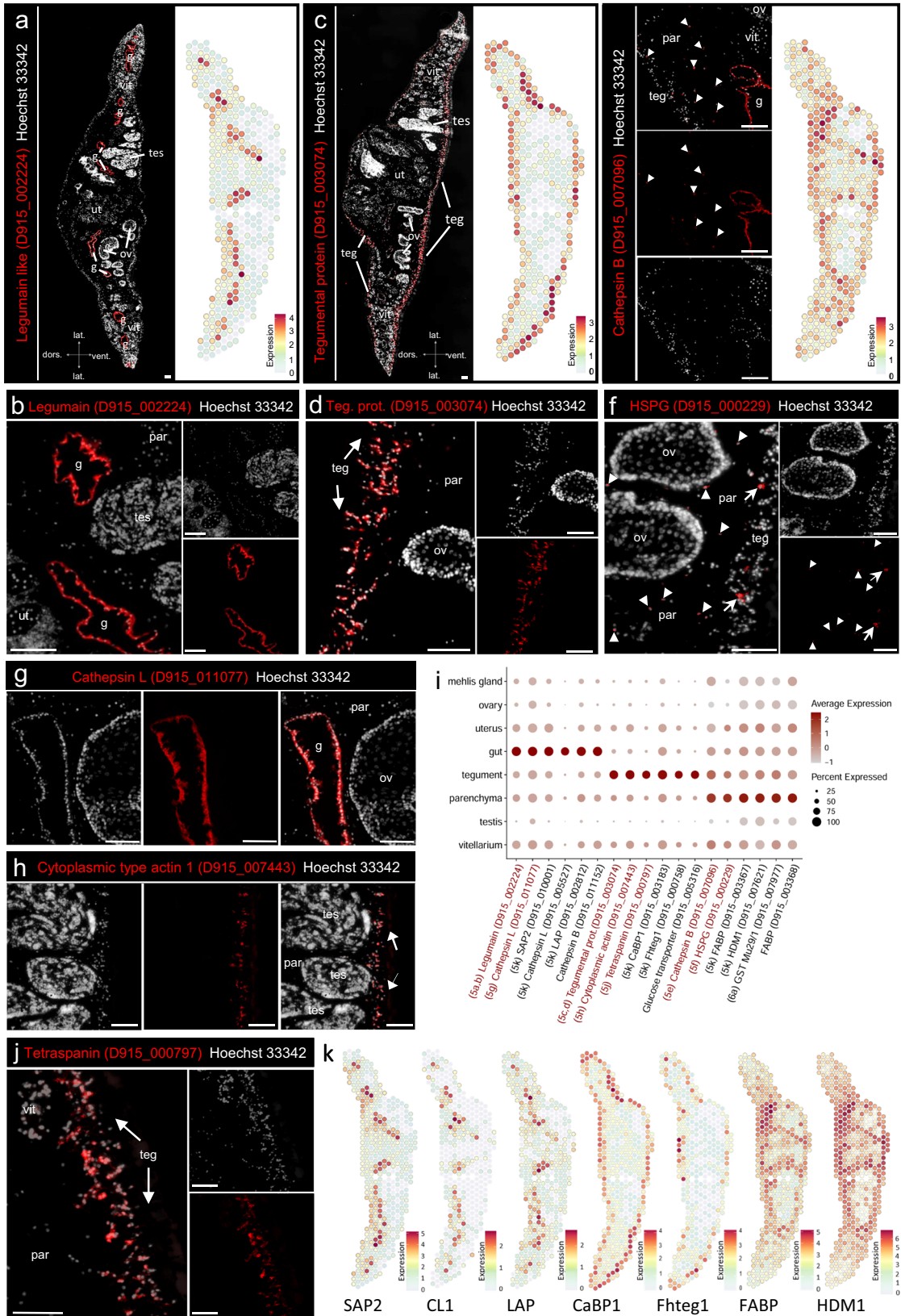

the different isoforms and revealed that almost all Mu-class GSTs have a common preference for parenchymal expression.

One exception was GST-Mu5 (D915_002901), which was mainly expressed within the tegument (Fig. 6a, c). The two Sigma-class GSTs present in the dataset clearly diverged in their spatial expression patterns. GST-S1 (D915_001201) expression in the vitellarium clearly

exceeded that in all other organs and thus matched the results of LaCourse et al.[46]. GST-S2 (D915_006844) was only recently identified using bioinformatics[42], but its localization and function were still unknown. We found that its expression was highest in the parenchyma and thus resembled the patterns we found for Mu-class GSTs. Diverging spatial expression patterns were also obvious for the two omega-

**Fig. 5 | Markers of liver fluke somatic tissues: gut, tegument and parenchyma.**
**a, c** Fluorescent in situ hybridization (FISH) overview (left) and spatial projection (right) of the gut marker legumain (D915_002224) (**a**) and a tegumental EF hand domain-containing protein (D915_003074) (**c**). Section orientation is indicated at the bottom (dors: dorsal, vent: ventral, lat: lateral). **b, d** Magnification of (**a**) and (**c**), respectively. **e** FISH (left) and spatial projection (right) of cathepsin B (D915_007096). This cathepsin B is expressed in the worm parenchyma as well as in the gut, white arrowheads indicating positive parenchymal cells. **a, c, e** Expression level encoded by color (gray = low, red = high). **f** FISH of basement membrane-specific heparan sulfate proteoglycan core protein (HSPG) (D915_0000229). HSPG is expressed in small cells within the worm's parenchyma (white arrowheads) as well as in larger cells below the tegument (white arrows). **g** FISH of cathepsin L (D915_011077). **h** FISH of cytoplasmic type actin 1 (D915_007443). **i** DotPlot showing expression profiles of selected tissue markers of the gut, tegument or parenchyma cluster. Dot color encodes the average expression level (mean of UMI counts, normalized and scaled) across all spots within a cluster. Dot size encodes the

percentage of spots within a cluster that have captured this transcript. Please note: While spatial plots (**a, c, e, k**) are shown for only one representative section, the DotPlot includes expression data from all four tissue sections in the dataset. Genes labeled in red were validated by FISH. Figure panels showing the respective FISH experiment and/or spatial projection are indicated. Source data are provided as a Source Data file. **j** FISH of a tegument-specific tetraspanin (D915_000797). **k** Spatial projections showing expression patterns of selected vaccine candidate genes: saposin-like family protein 2 (SAP2, D915_010001), cathepsin L1 (CL1, D915_005527), leucine aminopeptidase (LAP, D915_002812), tegumental calcium-binding protein CaBP1 (D915_003183), uncharacterized tegumental protein Fhteg1 (D915_000758), fatty acid-binding protein Fh15 (FABP, D915_003367), helminth-defense molecule 1 (HDM1/MF6, D915_007621). Expression level encoded by color (gray = low, red = high). **a–h, j** Scale = 100 μm. g: gut, ov: ovary, par: parenchyma, teg: tegument, tes: testis, ut: uterus, vit: vitellarium. For numbers of ISH experiments performed for each gene, see "Statistics and reproducibility" and Supplementary Data 9.

class GSTs. While GST-O1 (D915_001421) was mainly expressed in the vitellarium and uterus and somewhat less in the ovary, GST-O2 (D915_001777) was predominantly expressed in the parasite's tegument. These results support a role for GST-O1 within the reproductive system but suggest a different role for GST-O2 in protecting the parasite's barrier to the host. Finally, we found two microsomal GSTs with expression patterns complementing each other. While GST-m1 (D915_002950) was expressed in almost all tissues except testis, GST-m2 (D915_007840) showed strong expression within spots assigned to the testis cluster. In conclusion, most organs are characterized by a specific set of GSTs, which may reflect different needs of the tissues with respect to detoxification. Particularly noteworthy is the parenchyma, which expresses the greatest diversity of GSTs (Fig. 6c).

### The liver fluke gut and tegument express distinct ABC-B transporters

ABC transporters represent another interesting protein family of which some members help the parasite to defend against toxic products, most likely including drugs such as triclabendazole[5]. In addition to a possible role in TCBZ resistance[5], ABC transporters may be interesting targets for developing new treatments that enhance the efficacy of existing drugs or that interfere with the physiology of the parasite[47]. However, to date, there has been almost no information on the spatial expression of ABC transporters in this parasite that would provide first insight into their biological function. By phylogenetic modeling of 27 F. hepatica ABC transporter sequences together with human and C. elegans ABC transporters from all subfamilies (A, B, C, D, E, F, G), we verified in total: four ABC-A, twelve ABC-B, three ABC-C, two ABC-D, one ABC-E, two ABC-F and three ABC-G members (Supplementary Fig. 10b).

As subfamily B is particularly interesting with regard to drug resistance[5], we have focused on this subfamily in the further course. For seven family members, we identified an association to specific organs, while expression of the others was rather weak and dispersed (Fig. 6b, c, Supplementary Fig. 9b). Most striking was the expression of four genes (D915_007347 D915_006539, D915_007681 and D915_001064) in tissues at the host-parasite interface. Particularly noteworthy was the strong expression of D915_007347 in the tegument. D915_001064, on the other hand, was the only ABC-B transporter that was mainly expressed in the intestine. Interestingly, all four genes have in common that their proteins were localized in the membrane of extracellular vesicles (EV) in previous studies[48,49]. It has therefore been suggested that ABC-B transporters play a role in EV packing[49]. Our data now implies that different ABC-B transporter isoforms may be involved in EV formation depending on the organ of origin.

Several ABC-B genes have been discussed in terms of their potential role in drug resistance in F. hepatica. One (marker-

scaffold10x_157_pilon-snap-gene-0.179 = D915_004337) was located within the TCBZ resistance locus identified by Beesley et al.[50]. In our data, however, the gene was primarily associated with the testis (Fig. 6c, Supplementary Fig. 9b), suggesting a role in gametogenesis (e.g., protecting the germline against xenobiotics), rather than vital functions. The same holds true for D915_008893, which was also associated with the testis, and D915_007290, which was strongly expressed in the flukes' ovary (Fig. 6c, Supplementary Fig. 9b).

Since we and others demonstrated that TCBZ is taken up via the tegument[51,52], it would be reasonable to assume that mutation or increased expression of tegumental ABC transporters would favor TCBZ resistance. In our data, D915_007347 was found to be highly expressed in the tegument. A single nucleotide polymorphism (SNP) has been described for the same gene by Wilkinson et al.[53] for a small number of TCBZ-resistant F. hepatica, but could not be further confirmed as a resistance marker in following studies[5]. Therefore, it might be worthwhile to explore a possibly increased expression of tegumental D915_007347 in TCBZ-resistant strains as an alternative mode of resistance.

### Spatial transcriptomics aids in vaccine candidate prioritization

The liver fluke gut, tegument and parenchyma share a common feature that they are all in close interaction with the host's immune system, either by direct contact or by synthesis and release of excretory/secretory (ES) products and extracellular vesicles (EVs)[4,54]. This fits well with the fact that we detected several published vaccine candidates among the markers of these tissues in our data (Fig. 5i, k). Examples for the intestine were cathepsin L1 (D915_005527), leucine aminopeptidase (LAP, D915_002812) and the saposin-like family protein 2 (SAP2, D915_010001). Thus, in addition to their main digestive function, these enzymes also possess immunostimulatory effects[55–57]. The above-mentioned parenchymal FABPs and HDM-1 have also been the focus of vaccine studies, but have shown only moderate protection[58,59]. For the tegument, we would like to highlight two genes encoding the EF-hand domain-containing protein CaBP1 (D915_003183) and Fhteg1 (D915_000758). EF-hand domain-containing proteins have been investigated as vaccine candidates for schistosomes and F. gigantica and might also be worth exploring in F. hepatica. Fhteg1 is a tegumental protein with unknown function that has only recently been explored as a vaccine candidate for F. hepatica[60]. These findings encouraged us to search for potential vaccine candidates in our spatial transcriptomics dataset.

CD59-like proteins of the Ly6 family are thought to be involved in parasite-host interactions and have recently been proposed as vaccine candidates for Fasciola spp.[61] To be able to prioritize and select certain Ly6 proteins for vaccine trials, knowledge of the spatial expression of the different family members would be highly beneficial. Until now, only Ly6-Q (D915_008394) could be detected in the tegument

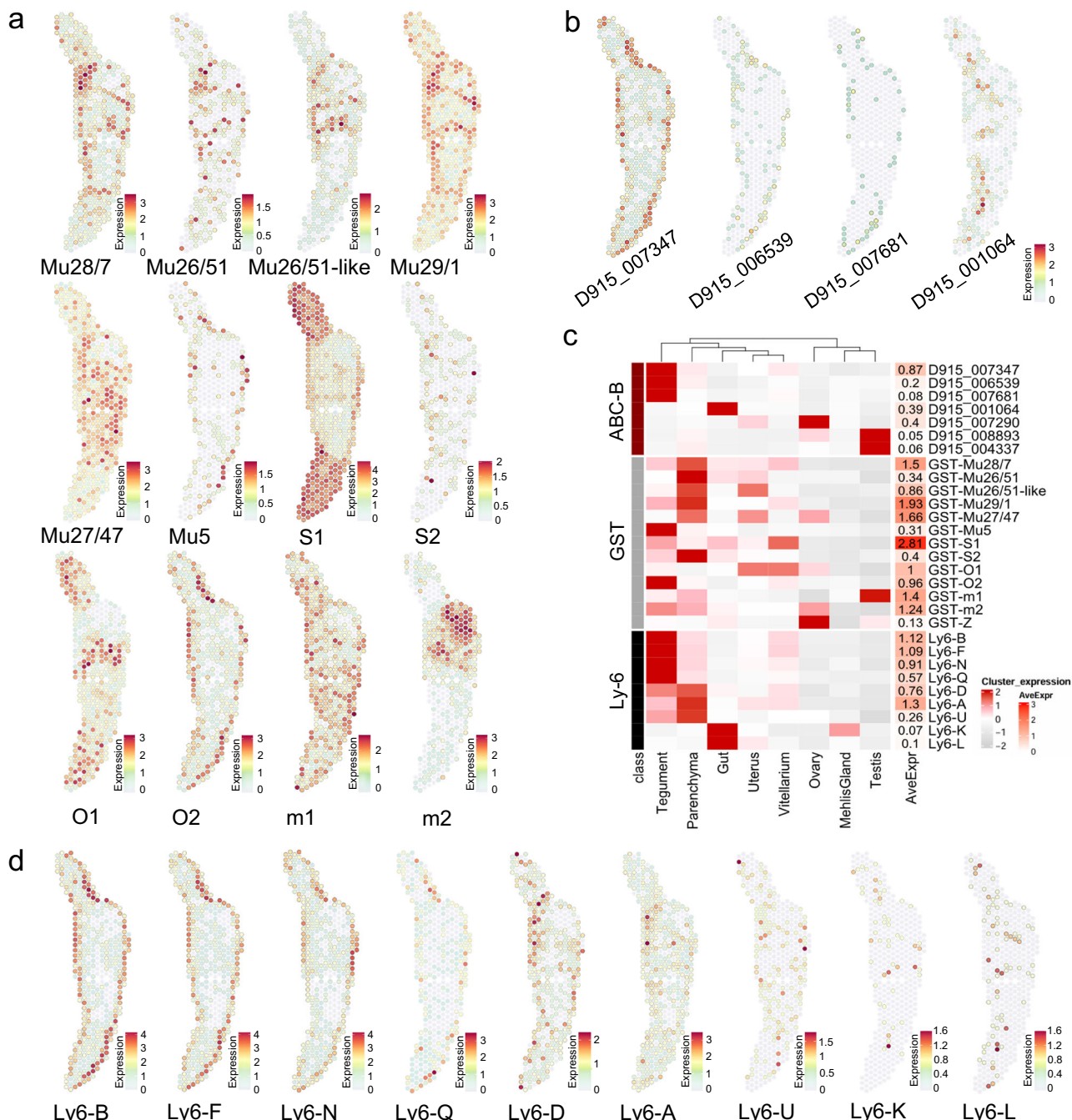

**Fig. 6 | Tissue-specific expression of detoxifying enzymes (glutathione S-transferases), efflux transporters (ABC transporters) and Ly6 proteins.**
**a** Spatial projections showing expression patterns of *F. hepatica* glutathione S-transferases (GSTs). Mu-class GSTs: Mu28/7 (D915_008526), Mu26/51 (D915_008966), Mu26/51-like (D915_008366), Mu29/1 (D915_007977), Mu27/47 (D915_010266), Mu5 (D915_002901); Sigma-class GSTs: S1 (D915_001201), S2 (D915_006844); Omega-class GSTs: O1 (D915_001421), O2 (D915_001777); and microsomal GSTs: m1 (D915_007840), m2 (D915_002950). Zeta class GST (D915_006391) is shown in Supplementary Fig. 9a. For GST phylogeny, see Supplementary Fig. 10a. **b** Spatial projection showing expression patterns of selected *F. hepatica* ABC-B transporters. See Supplementary Fig. 9b for spatial projections of ABC-B genes not shown in (**b**). For ABC-transporter phylogeny, see Supplementary Fig. 10b. **c** Left: Heatmap of GSTs, ABC-B transporters and Ly6 proteins showing the average expression per cluster (mean of UMI counts, normalized and scaled). Please note: While spatial plots (**a, b, d**) are shown for only one

representative section, the heatmap includes expression data from all four tissue sections in the dataset. Right: Heatmap showing the average spot expression of those genes in the whole dataset (log1p normalized counts). Source data are provided as a Source Data file. The two testis-associated ABC-B transporters (D915_008893 and D915_004337) and D915_007290, which is preferentially expressed in the ovary, are not shown in (**b**). See Supplementary Fig. 9b for their spatial expression patterns. **d** Spatial projections showing expression patterns of tegumental, parenchymal and potentially intestinal *F. hepatica* Ly6 proteins. Tegumental Ly6 proteins: Ly6-B (D915_008996), Ly6-F (D915_008863), Ly6-N (D915_007373), Ly6-Q (D915_008394). Ly6 proteins with prominent expression in the parenchyma: Ly6-D (D915_006706), Ly6-A (D915_001097), Ly6-U (D915_000333). Ly6 proteins expressed in the gut cluster: Ly6-K (D915_009743), Ly6-L (D915_008235). See Supplementary Fig. 9c for Ly6 proteins not shown in (**c–d**). **a, b, d** Expression level encoded by color (gray = low, red = high).

proteome of adult parasites[25]. We first performed WormBase ParaSite BLASTp to identify orthologues of *F. gigantica* Ly6 proteins within the *F. hepatica* genome (PRJNA179522). In total, we found 19 FhLy6 proteins (see Supplementary Data 6 for details). We then searched our spatial transcriptome for all family members. Thereby, we identified four tegumental Ly6 proteins (Ly6-B (D915_008996), Ly6-F (D915_008863), Ly6-N (D915_007373), Ly6-Q (D915_008394)), three mainly parenchymal Ly6 proteins (Ly6-A (D915_001097), Ly6-D (D915_006706), Ly6-U (D915_000333)) and two weakly expressed but potentially intestinal Ly6 proteins (Ly6-K (D915_009743), Ly6-L (D915_008235) (Fig. 6c, d). Three others (Ly6-M, Ly6-O, Ly6-R) were detected predominantly in the testis, an organ not vital for the parasite, and therefore appear not to be the best vaccine candidates (Supplementary Fig. 9c). Thus, spatial transcriptomics could extend our knowledge on expression of FhLy6 proteins enabling us to prioritize potential candidates for future vaccine trials.

### Prioritizing tegument- and gut-expressed genes for drug repurposing

With the rationale that new drug targets may be particularly found among tegumental or intestinal proteins (organs vital for the parasite), we sought to survey these clusters for putative targets. Therefore, we selected all genes in the dataset whose expression in the tegument or gut cluster was above the mean expression of these genes in all clusters. As a result, we obtained 474 and 246 genes for the tegument and intestine cluster, respectively (Fig. 7a). Next, we searched for orthologs of all eukaryote target proteins registered in ChEMBL for which there are small-molecule drugs in at least phase 3 of clinical development. Thereby, we found 11 potential targets of 24 drugs in the tegument and 10 targets of 37 drugs in the intestine (Fig. 7a, Supplementary Data 7). These included three ABC-B transporters (tegument: D915_007347 and D915_006539, gut: D915_001064), two solute carriers (gut: D915_004198 and D915_004176), two proteases (gut: D915_008045, D915_ 001479) and a chloride ion channel (tegument: D915_008739).

Based on our previous research on protein kinases of *F. hepatica*[6], we focused on this particular class of druggable proteins. There were two kinases among the predicted targets in our list of tegument-expressed genes (serine/threonine kinase, D915_002343; protein kinase C beta (PKCβ), D915_006901) and one for the gut (mitogen-activated protein kinase kinase kinase 15, D915_004118) (Supplementary Data 7). The *pkcβ* was one of four *pkc* genes that we identified in the genome of *F. hepatica* and the only one that showed notable expression in the tegument cluster (Fig. 7c, d). Based on an amino acid identity of 73.36% between the catalytic domains of *F. hepatica* and human PKCβ (Fig. 7b), we made use of the highly isoform-specific human PKCβ inhibitor ruboxistaurin (LY333531)[41] to test whether targeting of PKCβ causes anthelminthic effects. Indeed, treatment with 50 μM ruboxistaurin killed adult *F. hepatica* within 72 h of in vitro culture, a potency comparable to the gold standard triclabendazole (Fig. 7e, f, Supplementary Movies 1 and 2). This is one clear example of how the spatial transcriptome may help drug search and target prioritization.

## Discussion

The biology of parasitic metazoans, including the function of their various tissues, is still poorly understood. Their sheer size and cell number, complex life cycles, and a lack of molecular tools are turning parasite research into an experimental challenge. Nevertheless, the need for novel therapies against these pathogens has steadily driven research in this area. The increasing number of parasite genomes and the development of ground-breaking "omics" technologies in recent years have also opened up new opportunities for parasite research[4,62]. With our study, we are now able to provide a comprehensive 2D expression atlas of a multicellular parasite, the common liver fluke *F. hepatica*.

Various approaches have been used in the past to analyze the proteome or transcriptome of body regions, individual organs and tissues to gain molecular insights into the complex biology of these organisms. These approaches used different physical and enzymatic preparation techniques or laser capture microdissection (LCM). These classical methods, however, usually focused on a small number of selected organs. For example, the *F. hepatica* and schistosome tegument proteome were studied using different physical and enzymatic preparation techniques in order to detach the tegumental syncytium from the worm surface[25,63]. Furthermore, LCM technology has been used to excise and analyze the liver fluke gut and tegument proteome[48] as well as the transcriptome of *Schistosoma japonicum* intestine, vitellarium and ovary[64].

A game changer was the application of single-cell transcriptomics on helminths, to date applied for *S. mansoni* and *Brugia malayi*, which allowed to identify and characterize the cellular composition of parasites[28,29,65]. However, what scRNAseq cannot do is show the spatial arrangement of the cells' gene expression in the parasite. To do so, classical in situ hybridizations or immunohistochemistry are still necessary for every gene or protein of interest, which is associated with the time-consuming production and testing of transcript- or protein-specific riboprobes or antisera. In contrast, a spatial transcriptomics dataset, as presented here, allows to assess the whole spatial transcriptome of all tissues within a tissue section at once. And it does so without being forced to select individual organs of interest in advance, as required for methods such as LCM.

Spatial transcriptomics technologies can be broadly divided into two categories: imaging-based and sequencing-based technologies[66]. Imaging-based technologies offer high detection sensitivity and resolution but require a specific selection of targets for probe design and therefore some prior knowledge of the organism and a clear study hypothesis[66]. Since *F. hepatica* is still a relatively little-studied organism and the aim of our study was to generate a dataset that would serve as a basis for the formation of new hypotheses for several years to come, a sequencing-based approach that covers almost the entire transcriptome provided more value. Visium from 10x Genomics was the only sequencing-based technology available on the market when we performed our experiments. It captures any polyadenylated mRNA and is therefore applicable to a wide range of different organisms, including parasites. An additional strength of Visium is the availability of H&E staining of the same tissue section from which the transcriptome is generated. This greatly facilitated tissue assignment during the analysis.

Sounart et al.[15] were the first to apply Visium to a multicellular parasite, the filarial nematode *B. malayi*. They were able to transcriptionally distinguish four tissue regions in the posterior part of the worm. We now succeeded in providing an even more comprehensive spatial gene expression atlas of a metazoan parasite, covering gene expression data on eight distinct tissue types and, by means of subclustering, resolving selected cell types and developmental stages. Compared to the previous study, we applied ISH to confirm that our spatial transcriptomics data reflects the actual localization of transcripts within the parasite and is able to predict markers for different parasite tissues, a control that we consider important for first-of-its-kind studies. As a parasitic flatworm, *F. hepatica* belongs to a group of parasites evolutionarily unrelated to filarial nematodes, which further increases the scientific value of our dataset.

One of the biggest weaknesses of the Visium technology, however, is its resolution. Even though adult liver flukes are comparatively large parasites, the entire organism and its organs are very small compared to mammals, for which the method was developed. Many different cells and tissues come together in a small area. With a given spot size of 55 μm, we were able to distinguish eight distinct parasite tissues. But only in the case of Mehlis' gland and vitelline cells we were also able to successfully characterize specific cell types or stages. The

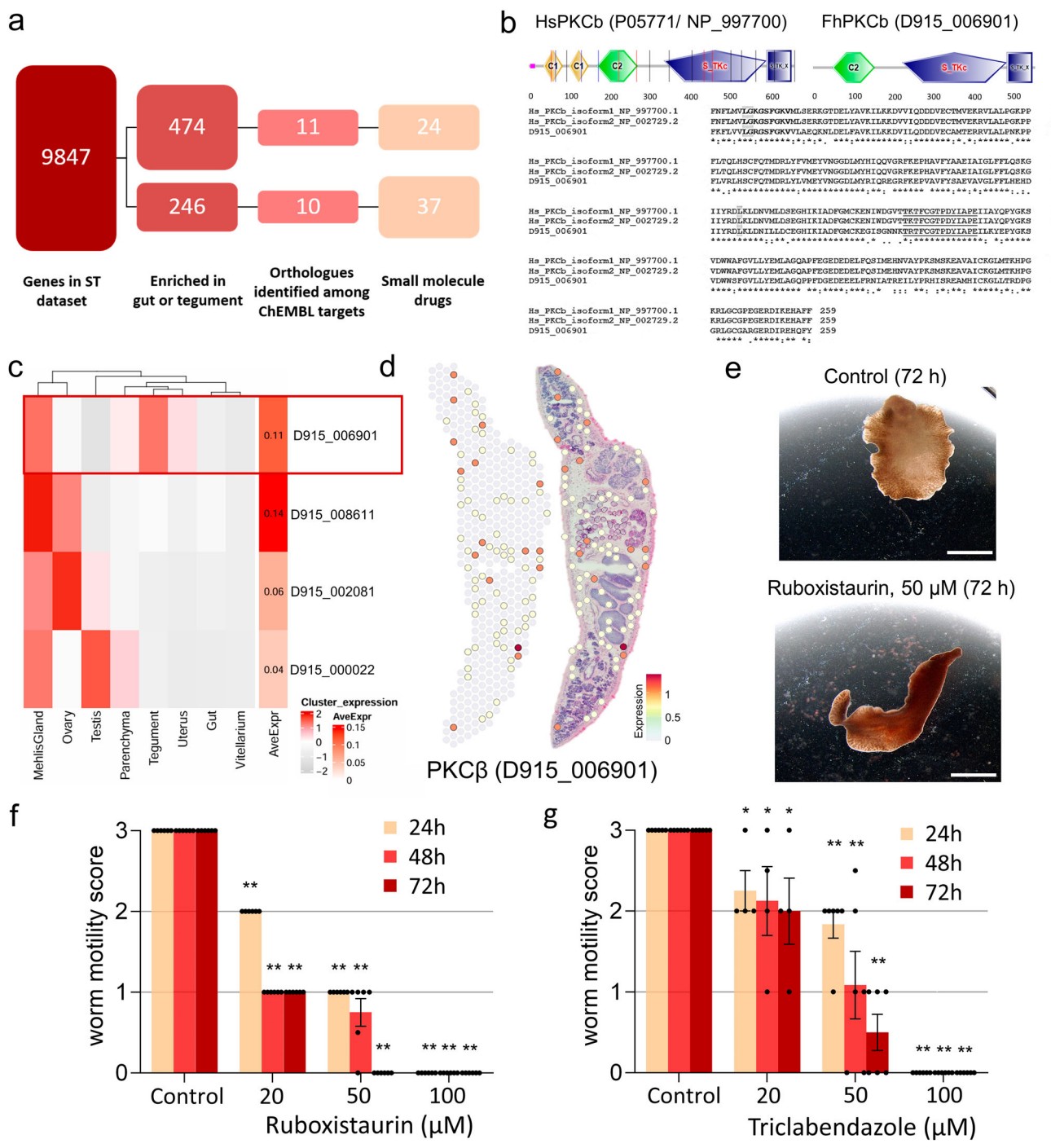

limited resolution also means that transcripts of two different tissue types, which are spatially close to each other, may merge in one spot. This needs to be considered when assigning genes to distinct tissues. We therefore recommend always performing a cross-check by assessing the spatial expression pattern, and not only relying on overview figures such as heatmaps or dot plots. If there is then still interest in a more precise or even (sub)cellular localization of individual transcripts, a method with higher resolution, e.g., ISH, is still necessary. In the future, we envision combining our spatial dataset with a comprehensive single-cell dataset of the adult liver fluke, so that we can benefit from both single-cell resolution and spatial information.

Due to the rapid development in the field, several more spatial transcriptomics technologies are commercially available today. Examples are Visium HD[67], Slide-seq[68]/Curio or Stereo-seq[69], which

now provide (near to) single-cell resolution. While Visium HD is reserved for mouse and human tissue samples, we clearly encourage the future use of Slideseq or Stereoseq in helminth research. A higher cellular resolution is especially beneficial for the analysis of smaller parasite life stages, e.g., immature liver flukes of just a few millimeters in size. Also, smaller structures, such as ganglia in the head region of adult liver flukes, may be covered. However, it should be noted that the already low detection efficiency of in situ capturing technologies decreases further with increasing resolution. This means that weakly expressed genes, such as many transcription factors, may not be captured reliably[66].

To test the power of our spatial atlas, we focused on genes of high interest in the field of parasitology: vaccine candidates and three gene families associated with drug action or resistance. Tegumental and

**Fig. 7 | A spatial transcriptomics-inspired drug repurposing approach. a** Flow chart showing gene numbers present in each step of our drug repurposing analysis. Eleven potential targets of 24 drugs were predicted in the tegument and ten targets of 37 drugs in the gut. **b** SMART analysis and alignment of catalytic domains of human and liver fluke PKCβ. The domain composition (C2 regulatory domain in front of the kinase domain) classifies both as conventional PKCs[89]. The catalytic domain of human PKCβ, which is bound by ruboxistaurin, shows 73.36% identity to the *Fasciola* ortholog. The conserved ATP binding site (bold) and the activation loop residues (underlined) are marked. The main residues involved in ruboxistaurin binding obtained for the human sequence[89] are Lys349, Gly350 and Lys467 (gray). **c** Left: Heatmap showing the average expression of different protein kinase C (PKC) genes per cluster (mean of UMI counts, normalized and scaled). Right: Heatmap showing the average spot expression of those genes in the whole dataset (log1p normalized counts). Red rectangle marks PKCβ with tegumental expression. Please note: While the spatial plot is shown for only one representative section, the heatmap includes expression data from all four tissue sections in the dataset.

Source data are provided as a Source Data file. **d** Spatial projection only (left) and overlay with H&E-stained tissue section (right) showing expression of PKCβ (D915_006901). Several positive spots can be seen along the tegument of the parasite. Mehlis' gland is not contained in this tissue section. For spatial projections of other PKCs, see Supplementary Fig. 9d. Expression level encoded by color (gray = low, red = high). **e–g** Adult flukes were treated for 72 h with different concentrations of the PCKβ isoform-specific inhibitor ruboxistaurin (20–100 μM) or triclabendazole as positive control. Motility was assessed every 24 h. Control worms were treated with the inhibitor solvent DMSO. Representative images are shown in (**e**) and motility scores for all time points and concentrations in (**f** and **g**) (score 3 = normal, 2 = reduced, 1 = severely reduced, 0 = no motility). See also Supplementary Movies 1 and 2. Data represent the mean ± SEM of $n = 4$ (triclabendazole at 20 μM) or $n = 6$ flukes (others) from 2 (triclabendazole at 20 μM) or 3 independent experiments (others) with 2 worms per condition and experiment. Significant differences to controls are indicated with $*p = 0.0333$ and $**p = 0.0022$ (two-sided Mann-Whitney U test). Scale bars correspond to 5 mm.

intestinal proteins appear as particularly attractive vaccine and drug targets thanks to their exposure to the host[4]. Thus, the spatial transcriptome, covering thousands of genes expressed within these tissues, can help prioritize candidate proteins in the future. As proof-of-concept, we identified a tegumentally expressed PKCβ and potent anthelminthic activity of a PKCβ inhibitor, ruboxistaurin. This compound was used in several clinical studies addressing diabetic retinopathy, was safe in patients and orally bioavailable[70]. This motivates follow-up studies on ruboxistaurin as drug candidate against *F. hepatica* infection. We are optimistic that even more fasciolicidal compounds can be discovered within the list of drugs revealed by our target prioritization strategy. Furthermore, we were able to visualize the spatial expression of 13 glutathione S-transferases, 12 ABC-B transporters and 16 Ly6 proteins, which revealed preferential expression of several members in particular organs, indicating site-specific gene functions.

These examples illustrate the utility of the dataset in exploring the spatial expression of a substantial number of genes. Thus, this dataset represents a valuable resource for both fundamental research and drug development against the common liver fluke. Insights into the spatial expression of genes will aid in deciphering their function and thus contribute to a better understanding of parasite biology. We envision these data to serve as a role model for the investigation of other understudied and experimentally challenging multicellular parasites, improving our understanding of their complex biology and facilitating the discovery of novel therapies for these pathogens.

## Methods
### Ethical statement
Animal experiments using rats (*Rattus norvegicus*) as model hosts were performed in accordance with Directive 2010/63/EU on the protection of animals used for scientific purposes and the German Animal Welfare Act. The experiments were approved by the Regional Council (Regierungspraesidium) Giessen (V54-19c20 15 h 02 GI 18/10 Nr. A16/2018). Animal welfare was monitored by assessing activity level, behavior, water and food intake, and fur and stool appearance of each animal.

### Preparation of parasites
To obtain adult *F. hepatica*, we orally infected male Wistar rats RjHan:WI (Janvier, France) at the age of 4–6 weeks with 25 metacercariae of an Italian parasite strain (Ridgeway Research, UK). Adult flukes were collected from the common bile duct at 12–14 weeks p.i. Worms were kept in RPMI 1640 (Gibco, Thermo Fisher Scientific, Germany) supplemented with 5% chicken serum (Gibco, Thermo Fisher Scientific) and 1% ABAM solution (c.c.pro, Germany) at 37 °C and 5% $CO_2$ for at least 1 h to allow clearance of gut contents. Parasites were then used for inhibitor treatments in vitro or embedded in O.C.T.

compound (Tissue-Tek, Sakura Finetek, Germany), frozen on dry ice and stored at −80 °C until use for Visium or in situ hybridizations.

### Cryosectioning and section placement
We performed spatial transcriptomics using the Visium Spatial Gene Expression Solution (10x Genomics, US), which makes use of probe-coated glass slides to capture mRNA from tissue sections. Adult *F. hepatica* embedded in O.C.T. compound were used to prepare transversal cryosections of 10 μm thickness with a cryostat Microm HM525 or Epredia CryoStar NX50 (Thermo Fisher Scientific). In order to identify the tissue region of interest when creating sections, we used a quick staining (Haema Quick Stain (Diff Quick), LT-SYS, Labor + Technik Eberhard Lehmann GmbH, Germany) to stain sections every 50–100 μm and checked them with brightfield microscopy. When a region of interest was reached, the consecutive section was transferred onto the Visium Spatial Gene Expression Slide. We finally used four cryosections from two different individuals (individual 1: capture area A & B, individual 2: capture area C & D). Sections were selected to cover ovary, testis and uterus at least in one of the two sections per individual. All four sections contained tegument, vitellarium, and gut tissue. Mehlis' gland was only present in section D. Slides with cryosections were transported on dry ice and stored in a sealed bag with desiccant at −80 °C until use.

### 10x Visium preparation and tissue optimization
To ensure good RNA quality of cryosections used in the spatial transcriptomics experiment, RNA quality of sections derived from the same tissue blocks was checked in advance. First, $3 × 10$ cryosections were collected in empty, prechilled Eppendorf tubes. Then, 300 μl 1x Monarch DNA/RNA Protection Reagent (New England Biolabs, Germany) was added, and the sample was stored at −80 °C until RNA isolation. For RNA isolation, the sample was thawed and refrozen three times in liquid nitrogen. RNA was then isolated using the Monarch Total RNA Miniprep Kit (New England Biolabs) following the manufacturer's instructions for homogenized tissue and lymphocytes. Concentration and integrity of isolated RNA were determined using the Bioanalyzer 2100 system (Agilent Technologies, Germany) and the Agilent RNA 6000 Pico kit (Agilent Technologies) according to the manufacturer's instructions (one electropherogram for each specimen used in the experiment is shown in Supplementary Fig. 1b).

To improve the efficiency of the Visium Spatial Gene Expression workflow, tissue permeabilization had to be optimized for liver fluke tissue in advance. This was done using the Visium Spatial Tissue Optimization Slide and Reagent Kit following the manufacturer's instructions (10x Genomics). In brief, *F. hepatica* cross sections were placed on a Visium Tissue Optimization slide, fixed and H&E-stained and imaged (Olympus IX81). Then, different tissue sections were permeabilized for different times (0, 3, 6, 12, 18, 24 and 30 min). A drop

(3 μl) of *F. hepatica* RNA (700 ng/μl) served as positive control. Released mRNA was captured by oligonucleotides on the Visium Tissue Optimization slide and subsequently reverse transcribed into fluorescently labeled cDNA. Fluorescent cDNA was then visualized with an Olympus IX81 microscope. The optimal permeabilization time for *F. hepatica* cross sections, resulting in maximum fluorescence signal with the lowest signal diffusion, was determined to be 12 min (Supplementary Fig. 1a).

## 10x Visium spatial gene expression workflow

We used the Visium Spatial Gene Expression Slide and Reagent Kit (10x Genomics) following the manufacturer's instructions with minor changes (for reagents, see Supplementary Data 8). Tissue sections were fixed in methanol, H&E-stained and imaged. Tile scanning was performed using a Leica DMi8 microscope (Leica Microsystems, Germany), equipped with a DMC2900 color camera (Leica Microsystems) and a HC PL APO 20x/0.80 objective (Leica Microsystems). Subsequently, the tissue was permeabilized for 12 min to release mRNA from the tissue. Tissue permeabilization was followed by reverse transcription, second-strand synthesis and denaturation. The cDNA from each capture area was then transferred to a corresponding PCR tube to allow amplification and library construction.

## 10x Visium library preparation and sequencing

We performed qRT-PCR to determine the optimal number of PCR cycles for cDNA amplification to generate a sufficient amount of cDNA for library construction (cycler: Rotor-Gene-Q, Qiagen, Germany, reaction volume: 20 μl, protocol modified: initial denaturation step at 98 °C for 3 min; 40 cycles of: 98 °C for 15 s, 63 °C for 20 s, 72 °C for 60 s). The cDNA of all samples was then amplified using 13 PCR cycles (Veriti 96-Well Thermal Cycler, Applied Biosystems, Thermo Fisher Scientific). After a cleanup (SPRIselect, Beckmann Coulter, Germany) as well as quality check and quantification (2100 Bioanalyzer, Agilent Technologies) (Supplementary Fig. 1b), the cDNA was sent for library construction and sequencing (Cologne Center for Genomics, University of Cologne, Germany). The libraries were sequenced on an Illumina Novaseq 6000 with a sequencing depth of about 100,000 reads per spot covered by tissue. All raw sequence data were deposited in the SRA under accession number PRJNA1047549.

## Mapping and quantification

Tissue and fiducial frames of each capture area were detected and aligned manually using the Visium Manual Alignment Wizard in Loupe Browser (v6.0.0) (10x Genomics). Sequencing data was then processed and mapped to the *F. hepatica* genome using the spaceranger count pipeline (v1.3.1) (10x Genomics). Our four sections covered a total number of 2020 spots, which captured a median of 2192 genes and 6138 UMI counts per spot.

In WormBase ParaSite 17, which we used as genomic resource for our analysis, two *F. hepatica* genomes were available: PRJEB25283[71] and PRJNA179522[16]. While the first genome provided the better genome assembly (BUSCO ASSEMBLY 70.1% vs. 65.6%) and was less fragmented (N50: 1.9 Mb vs. 161.1 kb), the gene annotation was slightly better in the case of the second genome (BUSCO ANNOTATION 69.9% vs. 71%). This had a marked impact on the mapping rate of our sequencing data to the transcriptomes (56% vs. 63% of the mean 118,604 reads per spot were confidently mapped to the transcriptome) and would probably further improve with improving genome quality. In addition, PRJNA179522 provided considerably more gene descriptions and thus much more biological information than PRJEB25283, in which many genes were only "NAs". Therefore, we chose PRJNA179522 as the basis for analyzing our spatial transcriptome data. The genome was slightly modified by adding the mitochondrial genome (Gene bank accession NC_002546.1). This version is available from the authors upon request.

## Data processing and clustering

Data from spaceranger was imported into Seurat (v4.3.0)[17,18] for further processing. First, a quality control was carried out by looking at the total number of UMIs and genes as well as the percentage of mitochondrial genes per spot (mitochondrial genes mostly below 5%; see Supplementary Fig. 2 for details). Manual filtering was applied by only including tissue-covered spots in the further analysis. Data was normalized and scaled using SCTransform with default parameters for each sample individually. Datasets were then merged using Seurat. On the merged dataset, we ran RunPCA for dimensionality reduction and RunHarmony (v0.1.1)[72] for batch correction (dims.use = 1:8, theta = 0, lambda = 4.7). The number of principal components for dimensionality reduction was determined by visual inspection of the ElbowPlot provided by Seurat. We then ran FindNeighbors (reduction = "harmony", dims = 1:8), FindClusters (resolution = 3) and RunUMAP (reduction = "harmony", dims = 1:8) to receive a first clustering. The clusters were then inspected and annotated according to the underlying tissue type (vitellarium, tegument, parenchyma, gut, uterus, testis, ovary and Mehlis' gland). For a small number of spots, the cluster ID did not correspond to the tissue seen in the histological image. Therefore, we exported barcodes with corresponding cluster IDs and UMAP coordinates from Seurat to Loupe Browser to manually correct mismatching cluster IDs of those spots. New cluster IDs were then re-imported into Seurat to proceed with marker gene analysis.

## Identification of marker genes

We used the FindAllMarkers function provided by Seurat to identify marker genes for each cluster (test.use ="roc", only.pos = TRUE) (Supplementary Data 2). Gene descriptions for all gene IDs were downloaded from WormBase ParaSite (BioMart) and then attached to the list of marker genes. Following the release of WBPS18, we adapted gene descriptions in our marker list according to this new version. Spatial expression patterns of marker genes were visualized using the SpatialFeaturePlot function in Seurat (crop = FALSE, pt.size.factor = 1, alpha = c(0.1,1), image.alpha = 0, stroke = 0.5). Additionally, the DotPlot function was used to provide an overview of the expression of multiple genes.

To identify marker genes for the two cell types of the Mehlis' gland, we manually selected spots covered by S1 and S2 cells in the Loupe Browser (Supplementary Fig. 6). Spots that were covered by other organs and tissues were labeled as "other." Barcodes and corresponding cluster IDs were then imported into Seurat to perform differential gene expression analysis. We used the FindMarkers function to identify marker genes for each cell type (test.use = "wilcox" (two-sided Wilcoxon rank sum test), assay = "SCT", only.pos = TRUE) (Supplementary Data 3). Annotations were added as described above.

## Gene ontology enrichment analysis

Gene Ontology (GO) annotation for *F. hepatica* was obtained from WormBase ParaSite (PRJNA179522, WBPS17) and added on by running InterProScan (v5.60.92.19)[73]. GO term enrichment analysis was performed using topGO (v2.46.0)[74]. Only marker genes with power values above the 25% percentile of each cluster were included in this analysis ("top 75%"). Analysis was done with weight01 method for all categories (BP and MF) with a node size restricted to ≥10. Significance was determined using a two-sided Fisher's exact test against all expressed genes.

## STRINGdb analysis

Molecular interactions were predicted using the STRING online tool (v11.5)[75] after uploading the *F. hepatica* proteome (PRJNA179522). GenIDs for the top 75% of marker genes of each cluster were retrieved from our marker list and entered as a multiple protein search. Default settings were used to predict interactions with a minimum interaction (confidence) score of 0.4 (medium level of confidence).

## Spatial co-expression analysis

To identify spatial gene expression patterns independent of Seurat clusters, we performed a spatial co-expression analysis with Giotto (v4.0.5)[30]. First, we created a new Giotto object containing gene expression matrix and cell locations from section C. Raw counts were then normalized with normalizeGiotto. Next, a spatial network (Delaunay triangulation network) was generated, to represent the spatial relationships between different spots. Using the BinSpect-kmeans algorithm, the 2500 most spatially coherent genes were identified, and a co-expression matrix of these genes was created using the detectSpatialCorFeats function. The results were visualized as heatmap (heatmSpatialCorFeats). Clustering resulted in 15 co-expression modules whose spatial expression patterns were summarized as metagenes and visualized using the spatCellPlot function. A correlation matrix of Giotto metagenes and Seurat clusters was calculated after normalizing metagene expression to a 0–1 scale. The results were visualized as heatmap (ComplexHeatmap v2.10.0)[76]. Finally, we extracted the gene composition of all metagenes and performed GO term enrichment analysis for metagene 1 & 5 in combination. GO term enrichment analysis was performed as described above with a node size restricted to ≥7.

## In silico investigation of *F. hepatica* gene families and cell cycle-associated genes

We used the keyword search in WormBase ParaSite to identify members of the *F. hepatica* β-tubulin, glutathione S-transferase, ABC transporter and PKC families within the *F. hepatica* genome (PRJNA179522). To compare and assign those sequences to known liver fluke or human sequences provided in NCBI, we used reciprocal BLAST searches with standard parameters. In addition, SMART[77] analysis was performed to confirm the domain structure of selected isoforms. *F. hepatica* orthologs of *F. gigantica* Ly6 proteins, recently characterized by Davey et al.[61], were identified by WormBase ParaSite BLASTp with standard parameters. Interspecies orthologs were defined as Ly6 proteins with >85% amino acid identity. A list of 108 stem cell- and cell cycle-associated genes was retrieved from Robb et al.[33]. Homologs in the PRJNA179522 genome were retrieved from WormBase ParaSite (BioMart). No homologs were found for four of these genes. Another 25 were not or so weakly captured in the dataset that tissue assignment was not possible. The remaining 79 genes were finally used for visualization.

Alignments of PKC-β catalytic domain sequences were produced in Clustal Omega (v1.2.4)[78]. For phylogenetic tree construction, GST and ABC transporter sequences were aligned using the MUSCLE alignment provided within the Molecular Evolutionary Genetics Analysis (MEGA, v10.2.4) software version X[79]. Phylogenetic trees were then constructed by the maximum likelihood method and JTT matrix-based model with 1000 bootstrap replicates using MEGA X.

To explore the spatial expression of gene family members, spatial plots were generated as described above. For better comparison of gene expression levels, we created a uniform color scale for some subsets of genes. Heatmaps were generated using the Complex-Heatmap package (v2.10.0)[76] after calculating the average expression of each gene per cluster. Gene expression values were centered and scaled for each gene individually.

## Drug target prediction

Tegument- and gut-expressed genes for our drug repurposing analysis were selected as follows: We first excluded all genes with very low expression levels by only selecting genes whose Log1p normalized average expression in all clusters was >1 (sum of all cluster values) and >0.25 in the respective cluster. Thereby, we retrieved 3329 and 2962 genes for the tegument and gut analysis, respectively. Next, the expression data was scaled to identify those genes that were enriched in either the tegument or the gut cluster. Only genes with a scaled expression >1 were used for the following drug target prediction (474 genes for the tegument, 246 genes for the gut).

A list of eukaryote single protein targets of approved drugs or drugs in phase 3 clinical trials was retrieved from the ChEMBL database (v34)[80]. *F. hepatica* protein orthologs were then identified by running BLAST+ (v2.13.0)[81] for the sequences of all 426 ChEMBL targets, employing an E-value cut-off of $10^{-5}$ and considering only the best matches. If the E-value was identical for two proteins (only the case for 0.0 results), both *F. hepatica* proteins were kept in the results list. In total, 218 *F. hepatica* proteins matched these criteria and were therefore considered potential drug targets. Among these, 11 and 10 targets were also present in our lists of tegument- and gut-enriched genes, respectively.

## Riboprobe synthesis

Templates for riboprobe synthesis were generated by TA-mediated cloning of pre-amplified cDNA sequences (Q5 High-Fidelity DNA Polymerase, M0491L, New England Biolabs & AccuPrime Taq DNA Polymerase, High Fidelity, 12346086, Thermo Fisher Scientific) with an average length of 400–500 bp. Primers for all genes can be found in Supplementary Data 9. The resulting PCR product was ligated (T4 Ligase, B0202S, New England Biolabs) with *AhdI*-digested (R0584L, New England Biolabs) pJC53.2[82] (26536, Addgene) and used to transform NEB® 10-beta competent *E. coli* (C3019H, New England Biolabs). Cloned sequences were confirmed by Sanger sequencing (Microsynth Seqlab, Germany) (sequencing primer: 5′-TTCTGCGGACTGGCTTTC-TAC-3′[83]) and WormBase ParaSite BLAST (see Supplementary Data 9 for results). Templates for in vitro transcription were generated by PCR amplification (Q5 High-Fidelity DNA Polymerase, New England Biolabs) from plasmids using a T7 extended primer (5′-CCTAA-TACGACTCACTATAGGGAG-3′[84]). In vitro transcription was finally performed using T3 or SP6 RNA polymerases (11031163001/ 11487671001, Roche, Germany) with the addition of Digoxigenin-11-UTP (11209256910, Roche).

## In situ hybridization

In situ hybridizations (ISH) (chromogenic (CISH) or fluorescent (FISH)) were performed as described earlier with slight modifications[85]. The samples used for ISH differed from those used in the ST experiments. The parasites were of the same age but originated from independent infections (i.e., from different host animals) and from different metacercariae batches (same Italian strain); 10 μm cryosections from adult *F. hepatica* were prepared as described above, post-fixed in 4% formaldehyde/PBS and permeabilized with PBSTx (0.5% Triton X-100). To inactivate endogenous peroxidase activity, slides were incubated in 0.03% $H_2O_2$/4x saline sodium citrate buffer (SSC) before continuing with prehybridization (FISH only). Hybridization reaction was carried out at 55 °C overnight. Probes were used at 1 μg/ml in hybridization buffer. The next day, multiple washing steps with hybridization washing buffer and decreasing concentrations of SSC were carried out, followed by blocking and antibody incubation (FISH: anti-DIG-POD, (11207733910, Roche), CISH: anti-DIG-AP, (11093274910, Roche)). After washing with maleic acid buffer (+0.1% Tween-20), the development reaction was carried out using the TSA Plus Cyanine 3 Kit (NEL704A001KT, Akoya Biosciences, US) (FISH) or BCIP/NBT (11383221001/11383213001, Roche) in alkaline phosphatase buffer (CISH). For FISH, tissue sections were counterstained with Hoechst 33342 (1 μg/ml) or DAPI (0.1 μg/ml) and mounted with ROTImount FluoCare (HP19.1, Carl Roth, Germany). For CISH, 80% glycerol was used for mounting.

Imaging of in situ hybridizations was performed on an Olympus IX81 microscope (Olympus, Japan) or a Leica DM IL microscope (Leica Microsystems). Fiji (ImageJ, v1.54f)[86] was used for linear adjustment of

brightness and contrast of acquired images. The adjustments were made individually for each fluorescence channel before they were merged.

### In vitro culture and inhibitor treatment

The anthelminthic activity of the PKCβ inhibitor ruboxistaurin (LY333531, S7663, Selleckchem) against adult stages of *F. hepatica* was assessed in vitro using different inhibitor concentrations (20, 50, or 100 μM). Worms were obtained as described above and cultured in RPMI medium supplemented with 5% chicken serum (16110082, Thermo Fisher Scientific) and 1% ABAM solution (10,000 units penicillin, 10 mg streptomycin, and 25 mg amphotericin B per ml, Z-18-M, c-c-pro, Germany) at 37 °C in a 5% $CO_2$ atmosphere for 72 h. Triclabendazole (32802, Sigma-Aldrich) at 20, 50, or 100 μM served as positive control and the solvent dimethyl sulfoxide (DMSO) equivalent to the highest drug concentration as negative control. Medium plus inhibitor was refreshed every 24 h. Inhibitor-induced effects on worm viability were assessed every 24 h using a stereo microscope at 10× magnification (M125 C, Leica, Germany) and the following scores: 3 (normal motility), 2 (reduced motility), 1 (minimal and sporadic movements), and 0 (dead).

### Statistics and reproducibility

The spatial transcriptomics workflow was performed once using one 10x Visium Spatial Gene Expression slide. No statistical method was used to predetermine sample size. We analyzed the spatial transcriptome of $n = 4$ tissue sections derived from two individual parasites (collected at different time points from different animals). Tissue sections were considered biological replicates. Tissue regions were sampled purposely from different body regions to include distinct organs of interest. Statistical analysis of RNA sequencing data was performed in R using Seurat, Giotto and topGO as described in previous method sections. Only spots covered by parasite tissues (selected manually) were included in the analysis. Data derived from empty spots was excluded. No further filtering was applied during the following analysis.

For in situ hybridization (Figs. 4, 5), we used $n = 2–5$ tissue sections in each independent experiment to confirm consistency in expression patterns. Parasite individuals and tissue sections were randomly allocated to different stainings. No statistical method was used to predetermine sample size. Replicates were performed on tissue sections deriving from distinct parasite individuals. Numbers of independent ISH experiments performed for each gene are listed in Supplementary Data 9.

For inhibitor testing, two worms were used per group and independent experiment. Inhibitor experiments were independently repeated 2–3x (technical replicates). Different parasite individuals were considered biological replicates. In total, we used $n = 4–6$ individual worms per condition. Worms harvested from rats were randomly allocated to either the inhibitor or the control group. Investigators were not blinded to allocation during experiments and outcome assessment. We used GraphPad Prism (Version 8) to process and analyze the worm-scoring data shown in Fig. 7. Statistical details are indicated in the figure legend (Fig. 7). No data were excluded from the analyses. Replications delivered reproducible results.

### Reporting summary

Further information on research design is available in the Nature Portfolio Reporting Summary linked to this article.

## Data availability

All raw sequence data was deposited in the NCBI Sequence Read Archive (SRA) under accession number PRJNA1047549. Filtered feature barcode matrices, slide images and manually corrected barcode-cluster assignments have been deposited at Zenodo (https://doi.org/10.5281/zenodo.10245261)[87]. Analyzed data can be visualized and explored in Cirrocumulus[88] using the following link: https://www.uni-giessen.de/haeberlein-lab/en/info. Please note: When uploading spatial data to Cirrocumulus, differential gene expression analysis is performed with scanpy and *t*-test by default. The list of tissue-specific markers displayed under "Sets" on the left and "RESULTS" on top is therefore very similar but not identical to our list in Supplementary Data 2 (calculated with ROC test in Seurat). Genes for which a "gene name" is specified in WormBase ParaSite, can only be found in Cirrocumulus via this ID, not by using their "D915"-ID (gene names for marker genes were included in Supplementary Data 2). Alternatively, a cloupe-file can be downloaded from Zenodo (https://doi.org/10.5281/zenodo.10245261)[87] to explore the data using LoupeBrowser (10x Genomics). The *F. hepatica* proteome (PRJNA179522), uploaded to the STRING database, is accessible via the organism identifier STRG0085JJO. Previously published accession codes used in this study include *F. hepatica* genome PRJNA179522 and *F. hepatica* mitochondrial genome NC_002546.1. Source data for graphs, DotPlots and Heatmaps are provided with this paper.

## Code availability

Code used for data analysis has been deposited at Zenodo (https://doi.org/10.5281/zenodo.10245261)[87].

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

## Acknowledgements

S.H. received financial support from the Deutsche Forschungsgemeinschaft (DFG) under grant HA 6963/2-1 and by the State of Hesse, LOEWE Center DRUID (LOEWE/1/10/519/03/03.001(0016)/53), which is gratefully acknowledged. O.P. received a scholarship from the Justus Liebig University Giessen. We would also like to thank Dr. Knut Beuerlein at the Rudolf Buchheim Institute of Pharmacology, JLU, for providing and supporting us with their Leica DMi8 microscope and thank the group of Prof. Bernhard Spengler (Institute of Inorganic and Analytical Chemistry, JLU) for access to their cryostat. The funders had no role in study design, data collection and analysis, decision to publish, or preparation of the manuscript.

## Author contributions

Conceptualization, S.G., S.H.; Methodology, S.G., O.P.; Investigation, S.G., O.P., T.S.; Visualization, S.G.; Resources, Z.L., O.P.; Writing—Original Draft Preparation, S.G.; Writing—Review & Editing, all authors; Supervision and Funding Acquisition, S.H.

## Funding

## Competing interests

The authors declare no competing interests.
