## [Peer Review File · Nature Communications]

REVIEWER COMMENTS

Reviewer #1 (Remarks to the Author):

The manuscript presented by Svenja Gramberg and colleagues presents for the first time a spatial atlas of the liver fluke *Fasciola Hepatica*. Firstly, I would like to commend the authors for providing such a concise and well-presented manuscript, with some novel insights derived from their dataset. I would also like to apologise for the time that took me to review this manuscript (personal circumstances precluded me from doing this sooner).

This is an incredible resource for the parasitology community, as it provides novel insights about the expression pattern of candidate genes of interest that could be the focus of much needed therapeutic applications. The comments below are geared towards increasing the impact of the findings presented here, and I hope the authors find it useful. In particular, I'd like to see more *in silico* predictions, given that there is still a wealth of knowledge yet to be found in this dataset. As it stands, the results are interesting but slightly shallow, so the application of additional bioinformatic tools to predict tissue interactions, etc, would be really informative. I appreciate this might not be possible, but certainly encourage the authors to explore this aspect.

Line 60: Given that this manuscript is likely to set a standard for others in the field trying to replicate these findings, or use similar methods for their own studies, I'd encourage the authors to include a supplementary figure highlighting the optimisation workflow. For instance, if possible, I'd like to see the permeabilization time course and the bioanalyzer profile of the resulting RNA/libraries, etc. This could be really informative to the scientific community

Line 80: Is it not surprising that none of the pathways identified in 1d and 1e for the reproductive tissue contain GO terms associated with this function (e.g., spermatogenesis, meiosis, etc)?

Line 84: Was the validation done with the same species collected for 10X Visium? If not, and although I highly appreciate the *in situ* hybridisation presented here, I think it would be beneficial to validate these findings on independent samples, just to make sure the transcriptomics results are reproducible independently. In other words, it would be important to separate technical from biological validation.

Line 107 and throughout: This dataset is impressive, but I think these results could benefit from additional *in silico* analyses using spatial interaction analyses (e.g., Giotto to infer potential spatially-resolved tissue interactions), that could ultimately offer additional novel (and interesting)

biological insights. Is this even possible with this dataset? I'd definitely encourage the authors to explore this possibility a bit more.

Line 229 onwards: I appreciate the use of the transcriptomics prediction to support drug discovery. Indeed, the use of the PKC β inhibitor Ruboxistaurin is a fascinating example. However, I'd encourage the authors to elaborate further on this aspect? What are the histological features of flukes exposed to low and/or high concentrations of this drug? What predictions/hypotheses can be independently tested based on the ST dataset (e.g., major effects on reproductive tissues, tegument damage?) This aspect could be developed a bit more as it is of major interest in the field. Histology analysis would certainly complement these studies.

Line 252: Did we know that these GST enzymes were localised in the fluke's parenchyma already? It is not clear from the text whether this is a novel finding or just a validation of the transcriptomics data with previously published work.

Line 369: The hypothesis presented here is fascinating and could be tested based on the ST results. I wonder if the authors could/should include this in their study? In situ hybridisation of treated/untreated flukes could certainly support this aspect and increase the impact of the study overall.

Reviewer #2 (Remarks to the Author):

The current study applies spatial transcriptomics to undertake characterisation of major transcriptional differences between and among tissues of adult *Fasciola hepatica*. As the authors note, *Fasciola* are important and zoonotic parasites that cause major losses within cattle, sheep and relate animal industries.

The manuscript as written is more of a technically focussed paper demonstrating application of the 10X Genomics Visium platform for spatial transcriptomics in a parasitic helminth. This work contrasts recent single-cell and spatial transcriptomic studies in various other helminth species that achieve a reportedly lower level of resolution. The authors then define a series of tissue specific marker transcripts and verify these using FISH probes. They use these data to explore transcript localisation of a number of putative vaccine targets and major antiparasitic drug targets based on current treatments for *Fasciola*. Finally they undertake some drug development work targeting a tegument transcribed kinase.

Overall the study is well executed. The methods and analysis are sound and the authors succeed in demonstrating that their approach allows tissue specific transcriptional profiling in this species.

However, I do have a number of comments that I think need to be address.

Major comment

1. Most notably, the primary impact of the study, based on the authors focus in the introduction and discussion is the utility of the 10X method to provide a deeper resolution of the tissue specific transcriptome of parasitic helminths. This is valuable. However, the authors do not leverage this increased depth of resolution in their study. They have noted that they have quantified tissue transcriptional profiles covering ~70% of the *Fasciola* transcriptome (i.e, many thousands of genes). But for the most part, the major result sections focus on a small number of marker genes for various tissues, and then some noteworthy candidate genes or gene families in various contexts. The only analysis of the overall dataset is presented in Figure 1, showing the ability for Seurat to separate out the tissue transcriptional profiles and the providing an overall summary of the transcripts by GO analysis. I think this is a missed opportunity and I am curious why the authors haven't delved into these data in more data (e.g., more thoroughly exploring and comparing the tissue specific transcriptomes they have identified). Do they not have the resolution? Is there some other reason not to do this? I think this could make better use of the large dataset being produced.

Minor comments

1. Fig 1- What is the orientation of the fluke and the dotmap representation in Fig 1b? Are we seeing a dorsal, ventral or lateral view? If lateral, is the dorsal surface to the left or right. This should be labelled or indicated in the legend. I suggest indicating this throughout the figures in the manuscript as well.

2. Fig 2 – I think it would be helpful to provide more detailed labelling of the anatomical features in the FISH images, particularly in panels b through h. There is also no obvious conservation in the order of the FISH panels and the spatial transcription of the corresponding genes in 2i. This makes attempting to comparing the FISH data to the transcriptional data very cumbersome. For most of the genes, I need to scan through the specific gene ids in Fig 2i and then try to find them in the panels. This would be much easier if the plot was sorted in the same order as the FISH panels, and possibly even labelled b-h in 2i for each of comparison. Using Cathepsin L (DP15_011077) in panel 2h as an example, this is the last FISH image in the figure, but the first column in the spatial transcription. Anatomical labelling would also help with interpreting the images. I see in 2i that Cathepsin L is most highly transcribed in gut tissue. Panel 2h only provides a quite zoomed in and cropped view of the FISH image. I assume the red staining corresponds to gut tissue, but I can't

really tell. Similarly for panel 2b. Is the red staining gut tissue? I think so, but with no labels and no additional context by panel in the figure legend, it's difficult to know. I can see more context is provided in the manuscript text itself, but this should be interpretable from the figure directly if possible.

3. Line 134 – What is meant by 'next to reproductive tissues' in this context? Given the authors are presenting spatial transcriptomic data, do they mean physically next to each other? I don't see get that impression from the data shown in panel 3a, but maybe I have misunderstood. Or do they mean transcription was most abundant in tegument, followed by reproductive organs? I thought the latter, but according to Fig 3b, enolase is most highly transcribed in ovarian, then tegumental tissues, but not in testis or uterine tissue at all) and Prx is similarly highly transcribed in tegumental and uterine tissue and, to a lesser extent, ovary and vitellarium, but again, not testis. Could the authors please clarify what they mean here?

4. Fig 4 – Similar comments here to those raised in Fig 2. I like the comparison between FISH and the spatial transcriptomic data, but this is quite difficult to do with the figure layout as it. Also, the figure legend says that the gene ids highlighted in red are supported by FISH data. However, the labels don't always match up. For example neither gene in red text in Fig 4f (i.e., D915_002718 or DP15_010963) in Fig 4f appear to be represented in a FISH panel. Please check this.

5. Lines 229-231 – How do the authors reconcile their assertion that tegumental and intestinal proteins are particularly attractive for drug target development against their data in fig 5 showing that known drug targets for flukes (i.e., b-tubulin isoforms) are mostly transcribed in reproductive tissues? Wouldn't prior success point towards looking for additional targets within these tissues? What is the background information supporting prioritising tegumental or intestinal proteins instead? I think this needs to be strengthened here.

6. Fig 5 – please also indicate that no Mehlis gland is visible in 5e

Overall I think the work is interesting, but the relatively limited exploration of the overall breadth of the data is a missed opportunity and I think would provide greater impact. I think providing these data as a community reference is a great and valuable resource, but probably not enough on its own for publication in a high impact journal. With a more detailed comparative analysis of the data, the authors may be able to make some quite interesting data to flesh out their current study.

Reviewer #3 (Remarks to the Author):

Gramberg and colleagues present the first spatial transcriptomics analysis of the parasitic flatworm, *Fasciola hepatica*. This parasite is an important zoonotic pathogen, affecting millions of people worldwide and is a major livestock pathogen. The authors define multiple tissue types, identifying marker genes with enriched cellular processes that seem broadly relevant to the tissue type. They then focus on key aspects of the parasites biology with very applied outcomes, including identifying vaccine and drug candidates, and demonstrating a genomics-led approach to rationally chose a highly effective repurposed drug based on gene expression location.

Spatial transcriptomics is a rapidly evolving field. Although there are a couple of other examples applied to parasites, this work is a clear and significant advance over others such that I am confident that this will be defined as the “first properly executed and informative” spatial transcriptomics example for parasites. The focus on drugs and vaccines in particular will be very exciting to the field, and will stimulate new avenues of research into other parasites of medical and veterinary importance. The manuscript is generally well written, and the figures compelling and informative. I have made some comments and suggestions below that I think will strengthen the manuscript and make it more broadly informative and accessible to its readers. A key point to address will be expanding on the strengths and limitations of the approach, especially as the data is proposed as an important resource and that this study will set an important precedent for future research using this technique.

Well done. I am looking forward to seeing a revised version.

Comments:

Line 14: “Introduction“

The structure of the introduction is a little unconventional.

It does read like a bit of a sales pitch for spatial transcriptomics, rather than there are important questions in *Fasciola* biology that are difficult to answer, but spatial transcriptomics might be able to give some insight

Make sure the rationale is clear

Line 32: “F. hepatica”

- normally , would write the full genus name at the start of a sentence
- This is a very short paragraph. Expanding on the lifecycle, and/or impact of infection, might be informative? What happens when people or animals are infected? Why is this an important organism to study?

Line 58: “adult stage of F. hepatica, the life stage causing chronic liver disease”

- Given Nature Comms is a generalist audience, having some more background on the parasite itself in the introduction would be helpful
- One thing worth considering here and throughout - you have looked at one life stage. There are a number of morphologically distinct life stages. The reader needs to know this. Further, somewhere, you should mention that to build this resource, more lifestages are needed etc.

Line 59: “we first optimized the tissue permeabilization time using the 10x Visium tissue optimization workflow”

- It is not clear what you optimised - it is not explained in the results, and there is very little information in the methods.
- Given this is the first convincing example of spatial transcriptomics in a parasite, it would be good to expand on this.

Line 62: “Supplementary Fig. 1a”

- I dont seem to be able to access the supplementary data...

Line 75: “cell types that are spatially close to each other were combined in one cluster”

- I think this needs a more explanation and potentially, some analysis
- Given a core message of this paper is that this spatial atlas is a “valuable tool for both basic and applied research” (I dont disagree here), how reliable are spots with mixed tissue signals?
- Are there other data which can validate spots which contain mixed vs pure signals, if such a thing can be determined? Ie, tissue specific RNAseq data, or key marker genes?

Line 78: “Gene Ontology (GO) enrichment analysis of marker genes”

- Given some of the uncertainty of the previous point, how robust are these GO terms analyses?
- Marker genes have not been defined or described - is this the right place to introduce this?
- There needs to be more detail, backed by data, defining marker genes. Specifically, defining how well tissues are defined by marker genes.

Line 84: “”The tissue-specificity of selected marker genes was confirmed”

- This seems incomplete and needs more detail, but of course, those details are to come in the following sections.
- Perhaps it is not needed here.

Line 86: “Marker genes reflect functional complexity of gut, tegument and parenchyma”

- I am not sure I understand the importance of this section - it does read like a few different threads were put together here that didnt really belong elsewhere.
- It is not clear what “Functional complexity” means - is it biologically complex, or is technically complex, ie noisy?
- However, I wanted to ask about “marker” genes - looking at Fig2i, many of the marker genes for a particular tissue are expressed widely. To what degree is this true, or due to tissues merging in spots? You explicitly say you have identified new marker genes, but if most are expressed everywhere to some degree, how can they be a marker?
- To what degree are genes previously defined to be associated with a particular tissue defined as a marker gene in these data?
- In Fig2i, what is “average expression”? Worth defining in the figure legend.

Line 158: “F. hepatica.In addition”

- Space needed before “In”

Line 211: “Beta-tubulins are of research interest as they are molecular targets of triclabendazole”

- Are all F. hepatic beta-tubulins susceptible to triclabendazole?

Line 214: “, we identified two new β -tubulin isoforms”

- Isoforms or genes?

Line 249: ”sequenes”

- sequences?

Line 282: “Particular noteworthy”

- Particularly noteworthy?

Line 290: “Discussion“

- A lot of the discussion was repetitive of the results, and I didnt feel like there was a critical discussion of the strengths and weaknesses of the approach. Given this is the first properly application of spatial on a parasite, this sets a benchmark for future studies.

- Some things to consider are the technology itself (visium vs other spatial technologies), optimisation, choice of sections, confidence in cluster / marker assignment, genome used, lifestages etc. Lots of factors can influence the outcome. Some are worth thinking about and perhaps expanding on.

- Technology-wise, there are other technologies with higher resolution - would this be useful?

- It wont detract from the impact and importance of the data by being a bit critical.

- Some of the reiteration of the results could be simplified - expand on the results in the results section - it will help explaining the results themselves. Then focus on the bigger issues in the discussion itself.

Line 322: “it allows a quick and easy assessment of the whole spatial transcriptome of a tissue section”

- Is it quick and easy, really?

Line 339: “proofs”

- Proves?

Line 343: “evolutionary”

- Evolutionarily?

Line 344: “among the marker genes highlighted in our study,”

- This is quite vague
- Are the marker genes identifying the same tissue types? Ie are they evolutionarily conserved?

Line 478: “optimal permeabilization time was determined to be 12 minutes”

- What is this based on? What does optimal mean?
- What other time points were tested?

Line 502: “Approximately 63% out of the mean 118,604 reads per spot were confidently mapped to the transcriptome”

- Worth pointing out this is a limitation of the genome, and that perhaps more biology will become apparent as the genome is improved?

Line 508: “No filters were applied”

- It is very common in single cell and spatial analyses to have filters to remove poor quality or low-complexity data. Did you not have any?
- To what degree would applying no filters introduce noise in the analyses?

Line 619: “experusing”

- experusing?

Line 624: “For statistical analysis of non-sequencing data, GraphPad Prism (Version 8) was used to process the data”

- Would be good to be clear where this is relevant, and what tests were used.

Line 625: “Statistical details are indicated in the figure legend (Fig. 4).”

- How does this refer to Fig 4?

Pg 38: "Materials and Correspondance"

- Correspondance > Correspondence

Figure 2 vs Figure 3

- Why have different heatmaps been used?

- I think it would be better to be consistent.

Reply to reviewer comments

for the manuscript “Spatial transcriptomics of a parasitic flatworm provides a molecular map of vaccine candidates, drug targets and drug resistance genes”

We thank the reviewers for their critical and helpful comments. We conducted two new types of analyses of the spatial expression data (applying Giotto for spatial co-expression, and an *in silico* drug target prediction), added one new main figure, and four new supplementary figures. We extensively revised the structure of the results and discussion section according to the reviewer suggestions.

As requested, we reply point by point (text in blue). Line numbers in our reply refer to the line numbers of the “tracked changes” version of the manuscript. In addition, a clean version is provided in the merged manuscript file.

Reviewer #1 (Remarks to the Author):

The manuscript presented by Svenja Gramberg and colleagues presents for the first time a spatial atlas of the liver fluke *Fasciola Hepatica*. Firstly, I would like to commend the authors for providing such a concise and well-presented manuscript, with some novel insights derived from their dataset. I would also like to apologise for the time that took me to review this manuscript (personal circumstances precluded me from doing this sooner).

This is an incredible resource for the parasitology community, as it provides novel insights about the expression pattern of candidate genes of interest that could be the focus of much needed therapeutic applications. The comments below are geared towards increasing the impact of the findings presented here, and I hope the authors find it useful. In particular, I'd like to see more *in silico* predictions, given that there is still a wealth of knowledge yet to be found in this dataset. As it stands, the results are interesting but slightly shallow, so the application of additional bioinformatic tools to predict tissue interactions, etc, would be really informative. I appreciate this might not be possible, but certainly encourage the authors to explore this aspect.

- We are grateful for the positive feedback and the idea to add further *in silico* predictions to gain even more insight from our data. We now added a spatial co-expression analysis using Giotto (see more details below). Furthermore, we included an *in silico* prediction of potentially druggable targets within prioritized tissue clusters. This revealed 11 and 10 potential targets in the tegument and the intestine, which could be linked to 24 and 37 different drugs listed in ChEMBL, among which was PKC β , which we had selected as target for *in vitro*-confirmation of drug effects against the parasite (lines 523ff.). The prediction data are added in Fig. 7.

Line 60: Given that this manuscript is likely to set a standard for others in the field trying to replicate these findings, or use similar methods for their own studies, I'd encourage the authors to include a supplementary figure highlighting the optimisation workflow. For instance, if possible, I'd like to see the permeabilization time course and the bioanalyzer profile of the resulting RNA/libraries, etc. This could be really informative to the scientific community

- We appreciate the interest in methodology details and have now included a supplementary figure, which shows the results of the tissue optimization and the Bioanalyzer profiles of RNA from cryosections and the cDNA library (Suppl. Fig. 1).

Line 80: Is it not surprising that none of the pathways identified in 1d and 1e for the reproductive tissue contain GO terms associated with this function (e.g., spermatogenesis, meiosis, etc.)?

- Thank you for this valid question. The basis of our GO term enrichment analysis was the gene ontology annotation provided in WormBase ParaSite (extended by running another InterPro scan). To better address your question, we checked the whole list of all WormBase GO terms for the whole *F. hepatica* proteome to see how many genes are associated with “spermatogenesis” and “meiosis” in general: There are only 2 genes for “sperm”/ “spermatogenesis” (D915_003117 & D915_004723) and 3 genes for “meiosis”/ “meiotic cell cycle” and “reciprocal meiotic recombination” (D915_003490, D915_007729, D915_010395). Both gene numbers (2 & 3 genes respectively) are lying below the “node size” threshold used in the analysis. Therefore, these terms were not included for enrichment analysis. From a biological point of view, it is of course to be expected that significantly more genes should be associated with these functions. Unfortunately, there is currently no database that provides this information.
- In case of meiosis, another point needs to be considered: The gene list used for GO term enrichment analysis was the list of “tissue markers” and not a list of all genes expressed in this tissue. A “marker” should be significantly more expressed in one tissue compared to all other tissues. This means for example that genes associated with “meiosis”, a process expected to occur in two tissues (ovary and testis), are probably not represented in the list of marker genes. Spatial co-expression analysis with Giotto (see also comment further below) allowed us to overcome the boundaries of individual clusters and to characterize common features across tissues, e.g. both gonads. Due to the above-mentioned limitation of the GO annotation, however, we could not cover the term meiosis there either. This is now briefly discussed in the corresponding results section (lines 182ff.).

Line 84: Was the validation done with the same species collected for 10X Visium? If not, and although I highly appreciate the in situ hybridisation presented here, I think it would be beneficial to validate these finds on independent samples, just to make sure the transcriptomics results are reproducible independently. In other words, it would be important to separate technical from biological validation.

- The validation was performed with different individuals than in the transcriptomics experiment. The parasites were obtained from independent infections (i.e. different host animals) and from different metacercariae batches. ISHs were therefore technical and biological validation in one. When selecting the sections for *in situ* hybridization, special care was taken to ensure that the section plane was similar in order to improve comparability.
- A statement clarifying this aspect was added to the methods section (lines 1032ff.).

Line 107 and throughout: This dataset is impressive, but I think these results could benefit from additional in silico analyses using spatial interaction analyses (e.g., Giotto to infer potential spatially-resolved tissue interactions), that could ultimately offer additional novel (and interesting) biological insights. Is this even possible with this dataset? I'd definitely encourage the authors to explore this possibility a bit more.

- We indeed had sought for deeper insights into spatial expression patterns within organs or to reveal overall expression gradients, e.g. involved in morphogenesis. However, we did not include many of them in the initial submission because of technical limitations we encountered. Among others we tried:
 1. **Uncovering transcriptional differences between the dorsal and ventral tegument** (dorso-ventral axis). Differences were found to be largely influenced by the proximity of the dorsal

and ventral tegument to other organs (matter of resolution), e.g. more “gonad”-related transcripts in ventral “tegument” spots, and therefore did not provide conclusive results.

2. **Subclustering of the gonads** to reveal expression patterns of different cell types and cell stages was hampered by the given resolution of the Visium approach combined with a quite heterogeneous cell arrangement in case of the testis and a relatively small organ size in case of the ovary. To discriminate different cell stages in these organs, an alternative technology with higher resolution (e.g. Curio seeker, Stereo seq) should be used in the future. Both were not available on the market when we performed our spatial transcriptomics experiment in January 2022. We added the use of alternative methods as a future prospect to our discussion (lines 697ff.).

- **Subclustering of the Mehlis’ gland** was successful: We managed to transcriptionally distinguish two different cell types in the Mehlis gland’ of the parasite (see Supplementary Figure 6). This was possible thanks to the well-demarcated position of these cells within the gland.
- Based on the reviewer’s suggestion, we felt motivated to try **Giotto** and obtained novel data, which we added to the manuscript. **Spatial-Co Expression analysis** succeeded in identifying subsets of genes with correlating spatial expression patterns. This enabled the identification of spatial gene expression patterns beyond the boundaries of individual clusters and allowed us to identify common features of ovary and testis. Shared genes thus appear to have a dual function in both tissues. Based on this analysis, we also analyzed and compared the tissue expression of a large number of cell cycle- and stem cell-relevant genes across different clusters (lines 182ff. and Fig. 4).
- Furthermore, we performed a **spatial interaction analysis with Giotto** to infer the effect of neighboring cell types on gene expression: The analysis pipeline basically worked (shown below: spatial network with Seurat clusters, cell proximity network, and cell proximity Barplot). However, since our spatial transcriptome data are not at a cellular resolution level but rather organ level, it did not yield meaningful or new interaction predictions. All spots within a given cluster showed a certain degree of homologous interactions (bar graph, in cyan color): Compact tissues, such as the gonads, more than spread-out tissues, such as the intestine and parenchyma. This was to be expected. A statistical frequency of heterologous interactions (between two different clusters) was only detected for intestine–vitellarium as well as intestine–parenchyma (red bars below). A direct interaction of intestine and vitellarium, however, is not to be expected, as these have no direct cell-cell contact but are separated by parenchyma cells (histologically-based finding). We hypothesize that the predicted spatial interaction is due to the fact that numerous intestinal cross-sections are located laterally in the worm section, where also the vitellarium is located, without Visium being able to dissolve the in between lying parenchyma. We therefore decided to not include this analysis to the revised manuscript.

Spatial interaction analysis by Giotto

Line 229 onwards: I appreciate the use of the transcriptomics prediction to support drug discovery. Indeed, the use of the PKC β inhibitor Ruboxistaurin is a fascinating example. However, I'd encourage the authors to elaborate further on this aspect? What are the histological features of flukes exposed to low and/or high concentrations of this drug? What predictions/hypotheses can be independently tested based on the ST dataset (e.g., major effects on reproductive tissues, tegument damage?) This aspect could be developed a bit more as it is of major interest in the field. Histology analysis would certainly complement these studies.

- Thank you for this excellent suggestion, additional histological data for ruboxistaurin-treated worms would indeed deliver further insights into ruboxistaurin action. Therefore, we repeated the ruboxistaurin treatment and fixed worms for histology when motility was significantly reduced (Score 1). This was after 24 h for worms exposed to 50 μ M and after 72 h in case of exposure to 20 μ M of ruboxistaurin. DMSO treated worms served as controls. We performed H&E stainings of cross-sections of 2-3 individuals per condition. We could not detect any structural differences between worms treated with ruboxistaurin and control worms at both concentrations. Therefore, the effect of the drug appears to be more subtle, primarily affecting cell function, but not directly tissue architecture (representative images of 1 individual per condition are shown below).

Line 252: Did we know that these GST enzymes were localised in the fluke's parenchyma already? It is not clear from the text whether this is a novel finding or just a validation of the transcriptomics data with previously published work.

- Our analysis added new knowledge to the existing data on GST expression within liver fluke tissues. GSTs have been reported to be widely distributed in liver fluke tissues, particularly in the parenchyma (doi: 10.1016/0014-4894(92)90142-w; doi: 10.1006/expr.1995.1098). However, localization of individual isoforms, especially Mu-class GSTs, was hampered by high sequence similarities and cross-reactivity of antisera. For instance, Creaney et al. (doi: 10.1006/expr.1995.1098) aimed at generating antisera specific for four FhGST-Mu isoenzymes (FhGST-Mu 29/1, 28/7, 27/47 & 26/51). However, only the anti-FhGST-Mu 29/1 was reactive and specific for this isotype, while anti-FhGST-Mu 26/51 was shown to be cross-reactive for all other isotypes.

- Unlike above traditional methodologies (IHC, ISH), Visium is a sequencing-based approach, so that even gene family members with highly conserved sequences are now distinguishable. Therefore, we were able to show for the first time that all GST-Mu isoforms except FhGST-Mu5 are preferentially expressed within the parenchyma. The relevant text passage has now been expanded to make this clearer (lines 342ff.).

Line 369: The hypothesis presented here is fascinating and could be tested based on the ST results. I wonder if the authors could/should include this in their study? In situ hybridisation of treated/untreated flukes could certainly support this aspect and increase the impact of the study overall.

- We believe this comment refers to line 396 (not 369): "... β -tubulin 2 and β -tubulin 3 as most likely candidates implicated in tegumental damage" [by triclabendazole]
- Please note: The tubulin chapter is no longer included in the main manuscript text of the revised version, as this topic no longer fitted into the revised manuscript structure and we also had to remove a few of the old passages from the manuscript due to word limitation. Spatial tubulin expression is now included in the supplement (Suppl. Fig. 7).
- If we understand the comment correctly, the reviewer would like to see an experiment in which *in situ* hybridization for the different β -tubulin isoforms is performed after triclabendazole treatment to see which isoforms are absent after treatment. The assumption would be that the cells in question stop β -tubulin expression or die because of the treatment so that no more transcript can be detected. To confirm the identity of the β -tubulin isoform(s) relevant for TCBZ action, we believe alternative strategies would deliver clearer results, such as heterologous expression and site-specific mutation to prevent TCBZ binding. This is an exciting field of research but a full topic for a separate research project.
- The proposed ISH to discriminate β -tubulin isoforms" is furthermore hampered by the high sequence similarity of the beta tubulins. The nucleic acid identity of 78-81% for most isoforms (beta-tubulin 5 and 6 still 62-67%), as can be seen in the percent identity matrix below, generated with Clustal Omega multiple sequence alignment. With a required RNA probe length of some hundred base pairs, we do not see the possibility of placing the probe in any section of the tubulin sequences without high probability of off-target effects.

1: D915_005076+D915_003963.beta-tub5	100.00	61.96	64.97	65.05	63.97	64.74	65.35	64.35	62.96
2: D915_008457.beta-tub6	61.96	100.00	66.44	66.29	65.92	66.89	66.44	66.06	66.06
3: D915_000542.beta-tub7	64.97	66.44	100.00	77.85	78.68	78.23	77.70	78.51	76.07
4: D915_007398.beta-tub1	65.05	66.29	77.85	100.00	79.13	81.12	80.97	78.96	79.46
5: D915_002311.beta-tub2	63.97	65.92	78.68	79.13	100.00	81.31	81.46	78.88	79.68
6: D915_004911.beta-tub3	64.74	66.89	78.23	81.12	81.31	100.00	95.58	79.41	79.76
7: D915_002077.beta-tub3	65.35	66.44	77.70	80.97	81.46	95.58	100.00	79.26	80.44
8: D915_000946.beta-tub8	64.35	66.06	78.51	78.96	78.88	79.41	79.26	100.00	80.77
9: D915_001342.beta-tub4	62.96	66.06	76.07	79.46	79.68	79.76	80.44	80.77	100.00

Reviewer #2 (Remarks to the Author):

The current study applies spatial transcriptomics to undertake characterisation of major transcriptional differences between and among tissues of adult *Fasciola hepatica*. As the authors note, *Fasciola* are important and zoonotic parasites that cause major losses within cattle, sheep and related animal industries.

The manuscript as written is more of a technically focussed paper demonstrating application of the 10X Genomics Visium platform for spatial transcriptomics in a parasitic helminth. This work contrasts recent single-cell and spatial transcriptomic studies in various other helminth species that achieve a reportedly lower level of resolution.

The authors then define a series of tissue specific marker transcripts and verify these using FISH probes. They use these data to explore transcript localisation of a number of putative vaccine targets and major antiparasitic drug targets based on current treatments for *Fasciola*. Finally they undertake some drug development work targeting a tegument transcribed kinase.

Overall the study is well executed. The methods and analysis are sound and the authors succeed in demonstrating that their approach allows tissue specific transcriptional profiling in this species. However, I do have a number of comments that I think need to be addressed.

Major comment

1. Most notably, the primary impact of the study, based on the authors' focus in the introduction and discussion is the utility of the 10X method to provide a deeper resolution of the tissue specific transcriptome of parasitic helminths. This is valuable. However, the authors do not leverage this increased depth of resolution in their study. They have noted that they have quantified tissue transcriptional profiles covering ~70% of the *Fasciola* transcriptome (i.e., many thousands of genes). But for the most part, the major result sections focus on a small number of marker genes for various tissues, and then some noteworthy candidate genes or gene families in various contexts. The only analysis of the overall dataset is presented in Figure 1, showing the ability for Seurat to separate out the tissue transcriptional profiles and the providing an overall summary of the transcripts by GO analysis. I think this is a missed opportunity and I am curious why the authors haven't delved into these data in more detail (e.g., more thoroughly exploring and comparing the tissue specific transcriptomes they have identified). Do they not have the resolution? Is there some other reason not to do this? I think this could make better use of the large dataset being produced.

- We are grateful for the feedback in order to increase the impact of the paper. Encouraged by your and another reviewer's comment, we performed more in-depth analyses with our spatial dataset. For detailed information, we would like to refer to our reply to a similar comment of reviewer 1 (see above: "Line 107 and throughout"). In brief:

1. We performed **spatial co-expression analysis using Giotto** to obtain information on characteristic gene expression patterns beyond the boundaries of individual clusters, which allowed us to identify common features shared between tissues. This analysis was added to the manuscript (lines 182ff. and Fig. 4).
2. We performed **spatial interaction analysis using Giotto** to infer the effect of neighboring cell types on gene expression. Knowledge gain from this analysis was limited, which is why we did not add any data to the manuscript.

3. We tried to **compare gene-expression patterns along body axes**, such as transcriptional differences between the dorsal and ventral tegument. This was not successful mainly because of technical limitations based on the given Visium resolution.
4. We included an ***in silico* prediction of potentially druggable targets** within prioritized tissue clusters. This revealed 11 and 10 potential targets in the tegument and the intestine, which could be linked to of 24 and 37 different drugs listed in ChEMBL (lines 523ff. and Fig. 7). This prediction was done earlier and guided us to the idea to repurpose the PKC β inhibitor ruboxistaurin against the parasites.

Minor comments

1. Fig 1- What is the orientation of the fluke and the dotmap representation in Fig 1b? Are we seeing a dorsal, ventral or lateral view? If lateral, is the dorsal surface to the left or right. This should be labelled or indicated in the legend. I suggest indicating this throughout the figures in the manuscript as well.

- Orientation and sectioning planes are now indicated. The same tissue section was used in all other figures throughout the main manuscript. Therefore, orientation is only given once in Figure 1.

2. Fig 2 – I think it would be helpful to provide more detailed labelling of the anatomical features in the FISH images, particularly in panels b through h. There is also no obvious conservation in the order of the FISH panels and the spatial transcription of the corresponding genes in 2i. This makes attempting to comparing the FISH data to the transcriptional data very cumbersome. For most of the genes, I need to scan through the specific gene ids in Fig 2i and then try to find them in the panels. This would be much easier if the plot was sorted in the same order as the FISH panels, and possibly even labelled b-h in 2i for each of comparison. Using Cathepsin L (DP15_011077) in panel 2h as an example, this is the last FISH image in the figure, but the first column in the spatial transcription. Anatomical labelling would also help with interpreting the images. I see in 2i that Cathepsin L is most highly transcribed in gut tissue. Panel 2h only provides a quite zoomed in and cropped view of the FISH image. I assume the red staining corresponds to gut tissue, but I can't really tell. Similarly for panel 2b. Is the red staining gut tissue? I think so, but with no labels and no additional context by panel in the figure legend, it's difficult to know. I can see more context is provided in the manuscript text itself, but this should be interpretable from the figure directly if possible.

- This is a valid point and we like to improve the arrangement and labeling of images to make the figures as easy as possible to understand for readers. A more detailed labelling was added to all FISH images.
- Genes in Fig. 2i (now 5i) were slightly reordered and panel labelling (reference to the figure panel where the corresponding FISH image can be found) was added to the DotPlot x-axis-labels. As a guidance for the reader, we already marked genes in all dot plots in red color if a corresponding FISH image is provided.

3. Line 134 – What is meant by 'next to reproductive tissues' in this context? Given the authors are presenting spatial transcriptomic data, do they mean physically next to each other? I don't see get that impression from the data shown in panel 3a, but maybe I have misunderstood. Or do they mean transcription was most abundant in tegument, followed by reproductive organs? I thought the latter, but according to Fig 3b, enolase is most highly transcribed in ovarian, then tegumental tissues, but not in testis or uterine tissue at all) and Prx is similarly highly transcribed in tegumental and uterine tissue and, to a lesser extent, ovary and vitellarium, but again, not testis. Could the authors please clarify what they mean here?

- Indeed the wording was not clear, we intended to say: expression was found in both, tegument and reproductive organs. We rearranged the whole chapter and peroxiredoxin and enolase were replaced with other examples of tegumentally expressed vaccine candidates. We hope the new version is free of further ambiguities (lines 297ff.).

4. Fig 4 – Similar comments here to those raised in Fig 2. I like the comparison between FISH and the spatial transcriptomic data, but this is quite difficult to do with the figure layout as it. Also, the figure legend says that the gene ids highlighted in red are supported by FISH data. However, the labels don't always match up. For example neither gene in red text in Fig 4f (i.e., D915_002718 or DP15_010963) in Fig 4f appear to be represented in a FISH panel. Please check this.

- A more detailed labelling of anatomic features was added to all FISH images.
- Genes in 4g (now 3g) were slightly reordered and a panel labelling was added to DotPlot x-axis-labels (reference to the figure panel where the corresponding FISH image can be found).
- Regarding “missing” FISHeS: Please note that FISH of D915_010963 was shown in Fig. 4e. ISH of D915_002718 was shown in Supplementary Figure 3.1c (as it was already indicated in the Figure legend of former Fig. 4). Now the whole “vitelline cell and egg” topic was moved to the supplement due to word limitations (Supplementary Fig. 5).

5. Lines 229-231 – How do the authors reconcile their assertion that tegumental and intestinal proteins are particularly attractive for drug target development against their data in fig 5 showing that known drug targets for flukes (i.e., b-tubulin isoforms) are mostly transcribed in reproductive tissues? Wouldn't prior success point towards looking for additional targets within these tissues? What is the background information supporting prioritising tegumental or intestinal proteins instead? I think this needs to be strengthened here.

- It is well described also for other flatworms, such as schistosomes, that drug targets can be expected to be found in tegument and intestine. We added some references to support this statement (lines 55ff.).
 - Zhu P et al. Advances in new target molecules against schistosomiasis: A comprehensive discussion of physiological structure and nutrient intake. *PLoS Pathog.* 2023;19(7):e1011498. doi: 10.1371/journal.ppat.1011498.
 - Mansour TE, Mansour JM. Targets in the Tegument of Flatworms. In: *Chemotherapeutic Targets in Parasites: Contemporary Strategies*. Cambridge University Press; 2002:189-214.
- Please note: A valid drug target can very well be expressed in other tissues, which we consider non-vital (reproductive) tissues, next to vital tissues. However, this does not mean that expression in the non-vital tissues is relevant for the killing activity.

6. Fig 5 – please also indicate that no Mehlis gland is visible in 5e

- Added as suggested (now Fig. 7e)

Overall I think the work is interesting, but the relatively limited exploration of the overall breadth of the data is a missed opportunity and I think would provide greater impact. I think providing these data as a community reference is a great and valuable resource, but probably not enough on its own for publication in a high impact journal. With a more detailed comparative analysis of the data, the authors may be able to make some quite interesting data to flesh out their current study.

- As addressed above, we succeeded in adding additional analyses which revealed shared gene expression profiles between tissues. We also show how to make use of such spatial gene

expression data in the context of drug target prediction. We believe this further increases the value of our manuscript for the research community. Information on organ expression of genes in a single, publicly available dataset is on its own a very valuable information gain and an important resource. We now also provide an access link for reviewers to check out our dataset in Cirro: https://opuckelw.github.io/spatial_fasciolomics. Within seconds, spatial expression of genes of interest can be visualized.

Reviewer #3 (Remarks to the Author)

Gramberg and colleagues present the first spatial transcriptomics analysis of the parasitic flatworm, *Fasciola hepatica*. This parasite is an important zoonotic pathogen, affecting millions of people worldwide and is a major livestock pathogen. The authors define multiple tissue types, identifying marker genes with enriched cellular processes that seem broadly relevant to the tissue type. They then focus on key aspects of the parasites biology with very applied outcomes, including identifying vaccine and drug candidates, and demonstrating a genomics-led approach to rationally chose a highly effective repurposed drug based on gene expression location.

Spatial transcriptomics is a rapidly evolving field. Although there are a couple of other examples applied to parasites, this work is a clear and significant advance over others such that I am confident that this will be defined as the “first properly executed and informative” spatial transcriptomics example for parasites. The focus on drugs and vaccines in particular will be very exciting to the field, and will stimulate new avenues of research into other parasites of medical and veterinary importance. The manuscript is generally well written, and the figures compelling and informative. I have made some comments and suggestions below that I think will strengthen the manuscript and make it more broadly informative and accessible to its readers. A key point to address will be expanding on the strengths and limitations of the approach, especially as the data is proposed as an important resource and that this study will set an important precedent for future research using this technique.

Well done. I am looking forward to seeing a revised version.

- We are very grateful for the appreciation of our work and like to further improve the manuscript according to the valuable suggestions. Amongst others, we re-structured the results and discussion part to allow readers an easier extraction of the key discussion points. We also added a more detailed discussion of limitations, alternative methods and future prospects, with which our manuscript can serve even better as a reference for future studies.

Comments:

Line 14: “Introduction”

The structure of the introduction is a little unconventional. It does read like a bit of a sales pitch for spatial transcriptomics, rather than there are important questions in *Fasciola* biology that are difficult to answer, but spatial transcriptomics might be able to give some insight

Make sure the rationale is clear

Line 32: “*F. hepatica*”

- normally , would write the full genus name at the start of a sentence

- This is a very short paragraph. Expanding on the lifecycle, and/or impact of infection, might be informative? What happens when people or animals are infected? Why is this an important organism to study?

Line 58: “adult stage of *F. hepatica*, the life stage causing chronic liver disease”

- Given Nature Comms is a generalist audience, having some more background on the parasite itself in the introduction would be helpful

- One thing worth considering here and throughout - you have looked at one life stage. There are a number of morphologically distinct life stages. The reader needs to know this. Further, somewhere, you should mention that to build this resource, more lifestages are needed etc.

- Many thanks for pointing out possibilities to improve the introduction. We restructured the introduction and added some more relevant background on the parasite for the general readership (lines 37ff.) as well as open questions with respect to its biology (lines 67ff.), which may be tackled by the use of spatial transcriptomics.
- The genus name in line 37 was revised as suggested

Line 59: “we first optimized the tissue permeabilization time using the 10x Visium tissue optimization workflow”

- It is not clear what you optimised - it is not explained in the results, and there is very little information in the methods.

- Given this is the first convincing example of spatial transcriptomics in a parasite, it would be good to expand on this.

- A valid point for which multiple reviewers demonstrated their interest. We have now included a section on preparation and tissue optimization in the methods section (lines 831ff.).
- We have also included a supplementary figure (Supplementary Fig. 1) showing the results of the tissue optimization.

Line 62: “Supplementary Fig. 1a”

- I dont seem to be able to access the supplementary data...

- Supplementary figures have been provided to Nature Communications as a pdf file along with all other manuscript files. We regret if the file was unavailable for reviewing and will check accessibility upon resubmission.

Line 75: “cell types that are spatially close to each other were combined in one cluster”

- I think this needs a more explanation and potentially, some analysis

- Given a core message of this paper is that this spatial atlas is a “valuable tool for both basic and applied research” (I dont disagree here), how reliable are spots with mixed tissue signals?

- Are there other data which can validate spots which contain mixed vs pure signals, if such a thing can be determined? Ie, tissue specific RNAseq data, or key marker genes?

- Since the Visium technology does not allow single-cell resolution, cell types covering a rather small area (such as tegumental cytons and subtegumental muscle cells) cannot be discriminated and were combined in one cluster (Supplementary Fig. 4) If the same cell type covers a larger area, such as for both gland cell-types of the Mehlis’ gland, cell-type specific clustering can be achieved (Supplementary Figure 6).
- There are various tools available that can be used to analyze the cell-type composition of Visium spots (so-called spot deconvolution). These tools, however, require additional input representing the gene signatures of known cell types, which are not sufficiently available for *F. hepatica*. Comprehensive lists of key marker genes to integrate into spatial data could be derived from:
 1. Tissue-specific RNAseq data: There are no gene lists published containing cell type- or tissue specific-marker genes from tissue-specific RNAseq analyses of *Fasciola spp.*
 2. Single cell RNAseq data: For other organisms, single-cell RNAseq data is commonly combined with spatial transcriptomics data to deconvolve the cellular composition of

tissue regions and to display the spatial arrangement of cell types. Unfortunately, until now there is no single-cell transcriptomics dataset that includes all relevant cell types and is therefore complete enough for such an analysis (including our own work). Tissues with difficult delimitation in spatial transcriptomics, such as the syncytial tegument, are still not accessible for the generation of single-cell data. But of course, we are aiming for such an integration of the two datasets in the future.

- We added a figure to demonstrate the resolution capability of Visium and better explain the presence of several cell types in one cluster (Fig. 1d, e). We now discuss the spatial limitation of Visium and mention alternative methods in our discussion (new RNA-capturing technologies such as Curio or StereoSeq, combinations of Visium and scRNAseq) (lines 682ff.).

Line 78: “Gene Ontology (GO) enrichment analysis of marker genes”

- Given some of the uncertainty of the previous point, how robust are these GO terms analyses?
- Marker genes have not been defined or described - is this the right place to introduce this?
- There needs to be more detail, backed by data, defining marker genes. Specifically, defining how well tissues are defined by marker genes.

- The order of the figures and with it the chronology of the results part has been changed. Marker genes and limitations of the dataset are now defined before the description of the GO term analysis (lines 111ff.)
- As shown by our ISH validation, the marker analysis itself already selects quite well for tissue-preferentially expressed genes. Moreover, since we only included the marker genes with the 75% best power values in the GO analysis (lines 945ff.), we consider it to be relatively robust.

Line 84: “The tissue-specificity of selected marker genes was confirmed”

- This seems incomplete and needs more detail, but of course, those details are to come in the following sections.

- Perhaps it is not needed here.

- The purpose of this first results section was to provide an overview of the entire dataset and our methodological approach to its analysis and also its biological validation via ISH. Therefore, we referred to the following figures that show the results of the ISH validation of selected marker genes: Fig. 2, 4, S3 in the previous version, now Fig. 3,5, S5,S6 in the revised version. The aspect of marker validation is now described in more detail (lines 111ff.).

Line 86: “Marker genes reflect functional complexity of gut, tegument and parenchyma”

- I am not sure I understand the importance of this section - it does read like a few different threads were put together here that didn't really belong elsewhere.

- It is not clear what “Functional complexity” means - is it biologically complex, or is technically complex, ie noisy?

- The main purpose of this section was to describe the clusters of somatic tissues and the selection of “marker genes” that were tested with ISH. We intended to condense the description of all clusters as good as possible to make the content more attractive to read also for non-parasitologists among the readership. Therefore, the three clusters (tegument, parenchyma, gut) were combined in one chapter. To add more content to this chapter and give it an overarching frame, we have now included our findings on known vaccine candidates, which are all mainly expressed in exactly these three tissues.
- “Functional complexity” referred to biologically complex. The title has been replaced (line 220).

- However, I wanted to ask about “marker” genes - looking at Fig2i, many of the marker genes for a particular tissue are expressed widely. To what degree is this true, or due to tissues merging in spots? You explicitly say you have identified new marker genes, but if most are expressed everywhere to some degree, how can they be a marker?

- Marker genes were determined by an accepted bioinformatics and statistic approach, which is the ROC test. This test identifies differentially expressed genes that ranked best to classify between the different tissues in the dataset. The ROC test returns a ‘classification power’ for each individual marker (ranging from 0 – no difference between groups/tissues, to 1 – perfect classification). Power values for each gene are contained in Supplementary Table S2.
- Based on our ISH experiments assessing expression patterns of several predicted marker genes, we can conclude that good statistical markers are most likely also good biological markers.
- Due to the limited resolution of Visium, however, it is likely that marker transcripts for one tissue were also captured to a limited degree by spots assigned to another tissue. As a result, a DotPlot or heatmap for marker genes from Visium data will rarely look as clear as a single-cell dataset, where individual cells are actually compared with each other. We realize that the interpretation of such DotPlots may be confusing to readers and therefore added this limitation to the text (line 688ff.).

- To what degree are genes previously defined to be associated with a particular tissue defined as a marker gene in these data?

- Several genes known to be characteristic for a particular tissue from literature were also found as markers in our data. We added this point, including examples of genes, to the results part (lines 121ff.).

- In Fig2i, what is “average expression”? Worth defining in the figure legend.

- Revised as suggested.
- Average expression is the mean expression of a certain gene in all spots of a cluster. This information was added to the legend.
- For your interest, the way of calculation:
 - Raw counts were corrected for technical variance and normalized during preprocessing using SCTtransform in Seurat.
 - Corrected counts are then used to calculate average expression
 - For visualization, Average Expression is Log1p normalized and scaled (z-score like), so that even different genes with large expression differences can be displayed in one plot.

Line 158: “F. hepatica. In addition”

- Space needed before “In”

- Revised as suggested

Line 211: “Beta-tubulins are of research interest as they are molecular targets of triclabendazole”

- Are all F. hepatic beta-tubulins susceptible to triclabendazole?

- To our knowledge, this isn’t resolved yet. We are not aware of any study that assessed binding affinities of different beta tubulins to triclabendazole.

- Chambers et al. (<https://doi.org/10.1007/s00436-010-1997-5>) performed a comparative experiment for albendazole, finding that the $\beta 2$ tubulin of *F. hepatica* showed strongest interaction.
- There is however evidence that TCBZ might bind at another site of the tubulin molecule than classical benzimidazole compounds, which might explain why *Fasciola spp.* are susceptible for triclabendazole, in contrast to many other worms species.
- Olivares-Ferretti et al. (<https://doi.org/10.1007/s11686-023-00692-z>) performed docking analyses of triclabendazole into *in silico* models of different *F. hepatica* tubulins. They suggested that TCBZ might bind the nucleotide binding site, rather than the benzimidazole binding site. Experimental proof is still lacking.
- Please note: The tubulin chapter is no longer included in the revised main part of the manuscript, as this topic no longer fitted into the revised structure and we also had to remove a few of the old passages from the manuscript due to word limitation. Spatial tubulin expression is now included in the supplement (Supplementary Fig. 7). See also our answer to the last comment of Reviewer 1 regarding experimental approaches towards identification of tubulins involved in triclabendazole action.

Line 214: “, we identified two new β -tubulin isoforms”

- Isoforms or genes?

- A small but not simple linguistic question, where the research community seems to have different practices. For several large gene families, such as tubulins or GSTs, the terms "isotype" and "isoform" are used interchangeably in literature (e.g. Stuart et al. 2021 (<https://doi.org/10.1007/s00436-021-07055-5>)). In the tubulin literature, the term "isotypes" is mostly in use (e.g. Ryan et al. 2008 (<https://doi.org/10.1016/j.molbiopara.2008.02.001>), Cwicklinski et al. 2015 (<https://doi.org/10.1186/s13059-015-0632-2>)). However, to our knowledge, the term "isotype" is mainly used in immunology to refer to antibody classes. We therefore decided to use the term “isoform” according to the definition below, to refer to gene products (in our case mRNA) derived from different but similar genes:
 - Protein Isoforms [NIH MeSH Terms definition]: “Different forms of a protein that may be produced from different genes, or from the same gene by alternative splicing”

Line 249: “sequenes”

- sequences?

- Revised as suggested

Line 282: “Particular noteworthy”

- Particularly noteworthy?

- Revised as suggested

Line 290: “Discussion”

- A lot of the discussion was repetitive of the results, and I didnt feel like there was a critical discussion of the strengths and weaknesses of the approach. Given this is the first properly application of spatial on a parasite, this sets a benchmark for future studies.

- Some things to consider are the technology itself (visium vs other spatial technologies), optimisation, choice of sections, confidence in cluster / marker assignment, genome used, lifestages etc. Lots of factors can influence the outcome. Some are worth thinking about and perhaps expanding on.

- Technology-wise, there are other technologies with higher resolution - would this be useful?
- It won't detract from the impact and importance of the data by being a bit critical.
- Some of the reiteration of the results could be simplified - expand on the results in the results section - it will help explaining the results themselves. Then focus on the bigger issues in the discussion itself.

- The suggestions to improve the value of the discussion are very appreciated. The discussion was largely revised following the suggestions above. We now included reasons for the choice of the Visium technology, its strength and weaknesses, associated limitations with regard to confidence in cluster/marker assignment and suggestions for future studies.
- Considerations that led to the choice of the genome and the sectioning plane were now included in the methods part (lines 890ff.).
- The parts related to individual gene families were removed from the discussion as suggested and incorporated into the results section as background information in order to reduce redundancies.

Line 322: "it allows a quick and easy assessment of the whole spatial transcriptome of a tissue section"

- Is it quick and easy, really?

- Once the dataset is prepared: yes it is. We now provide an access link for reviewers to check out our dataset in Cirro: https://opuckelw.github.io/spatial_fasciolomics/ Within seconds, spatial expression of genes of interest can be visualized.
- The statement has been removed to avoid misunderstandings.

Line 339: "proofs"

- Proves?

- Sentence was rephrased (line 670 ff.).

Line 343: "evolutionary"

- Evolutionarily?

- Revised as suggested, now included into the results (line 126).

Line 344: "among the marker genes highlighted in our study,"

- This is quite vague

- Are the marker genes identifying the same tissue types? Or are they evolutionarily conserved?

- Yes they are conserved for the same tissue types. We rephrased the sentence to clarify this even more (line 123ff.).

Line 478: "optimal permeabilization time was determined to be 12 minutes"

- What is this based on? What does optimal mean?

- What other time points were tested?

- A paragraph on tissue optimization has now been added to the methods section, which aims to answer all of the above questions (lines 831ff.).
- A supplementary figure displaying the results of the tissue optimization workflow was added as well (Suppl. Fig. 1).

Line 502: "Approximately 63% out of the mean 118,604 reads per spot were confidently mapped to the transcriptome"

- Worth pointing out this is a limitation of the genome, and that perhaps more biology will become apparent as the genome is improved?

- The achieved 63% is indeed somewhat lower than for human and murine datasets provided by 10x Genomics (~77-88%). As suggested, this is probably due to the limitations of the genome assembly and annotation of the available *F. hepatica* genomes and would likely improve with improving genome quality. This background information has been added (lines 890ff).

Line 508: "No filters were applied"

- It is very common in single cell and spatial analyses to have filters to remove poor quality or low-complexity data. Did you not have any?

- To what degree would applying no filters introduce noise in the analyses?

- Thank you for this question. Our decision not to use filters was based on the following consideration: The analysis pipeline used (Seurat) was originally developed for single-cell data. But spatial and single-cell datasets have different pre-conditions.
 - Single cell:
 - In a single-cell experiment, the material from which the sequencing data is derived from is often very heterogeneous. Cells are usually stressed by the dissociation process and even if a FACS sort for living cells is performed ahead of the barcoding step, cells can still die in the further course of the experiment. Furthermore, technologies based on the encapsulation of cells in oil droplets (e.g. 10x chromium) may deliver empty droplets or droplets with several cells (doublets/multiplets). In order to exclude affected cells, empty droplets, and multipliers from further analysis and thus reduce the noise, filtering is mandatory in single-cell experiments. This is achieved by setting thresholds, e.g. for the percentage of mitochondrial genes in the total gene count and the presence of an overall aberrantly high gene count.
 - Spatial:
 - In an ST experiment, fresh frozen tissue is used. Viable worms were fixed the same day they were isolated from the animal, without much of manipulation. High mitochondrial counts from cell stress or death are therefore not expected. Furthermore, forming of doublets/multipliers is per definition not possible because capture of transcripts occurs on an array of spots that are fixed on the slide.
 - False-positive data may derive from spots that are actually not covered by tissue, but might still capture some transcripts that have arrived there via diffusion (equivalent to empty droplets in single-cell experiments). In contrast to single-cell experiments, one can actually SEE where those "empty" spots are. Spots not covered by tissue were manually selected and excluded from the analysis. One could call this a kind of filtering.
- We realize that more background information may be of interest to readers. We now made some additions to the methods part:
 - (1) An explicit statement that a manual filtering was applied by excluding spots from the analysis that are not covered by tissue (previously just summarized as "tissue-covered spots were included in the further analysis") (line 910ff.).
 - (2) We had already assessed both quality parameters usually used for single-cell experiments (gene/UMI count, as well as mitochondrial percentage) and the results were provided in the original submission within Supplementary Figure 1 (Now Supplementary Figure 2). The vast majority of spots had a mitochondrial percentage below 5%. We regret that the supplementary data were unavailable to the reviewers during the reviewing process. When

we refer to Supplementary Fig. 2 in the methods, we now added the actual result of quality control (lines 909ff.).

Line 619: “experusing”

- experusing?

- Corrected to “using” (line 1062)

Line 624: “For statistical analysis of non-sequencing data, GraphPad Prism (Version 8) was used to process the data”

- Would be good to be clear where this is relevant, and what tests were used.

- “Non-sequencing data” refers to worm-scoring data, which was only shown in Figure 4 (now Figure 7). Therefore, we indicated statistical details (tests, sample number...) in the legend of this figure, and refer to this legend in the statistics section.
- As this sentence seems not to be phrased well, we rephrased as follows: “We used GraphPad Prism (Version 8) to process and analyze the worm-scoring data shown in Fig.7. Statistical details are indicated in the figure legend (Fig. 7).”

Line 625: “Statistical details are indicated in the figure legend (Fig. 4).”

- How does this refer to Fig 4?

- See answer above

Pg 38: “Materials and Correspondance”

- Correspondance > Correspondence

- Revised as suggested

Figure 2 vs Figure 3

- Why have different heatmaps been used?

- I think it would be better to be consistent.

- While the heatmaps only contain information on gene expression levels, the DotPlot also includes the percentage of spots in a cluster that have captured a particular gene. Thus, both visualizations are not exactly the same.
- Information on “percentage expression” from DotPlots can be very helpful in interpreting the data. However, it also means that for genes that were only detected in relatively few spots, the spots in the DotPlot become so small that they can hardly be recognized, when visualized together with genes with higher expression levels and broader detection. For this reason, heat maps and not DotPlots were used for visualization of gene families and functional gene lists. Therefore, we would prefer to keep both types of data visualization.

REVIEWER COMMENTS

Reviewer #1 (Remarks to the Author):

I have now read the authors' comments and appreciate the time they have taken to address my questions. The current version is a more robust manuscript and I hope the authors found my suggestions helpful. I am happy to support this manuscript for publication and I look forward to seeing this study in print.

Best wishes and good luck in your future studies!

Reviewer #2 (Remarks to the Author):

Typo line 56 should be "drug and vaccine targets"

Lines 67-71: Do the authors come back to these questions in their paper?

Lines 81-84: I think it would be better for this sentence to focus on tangible outcomes of the study.

Introduction: it's ultimately up to the authors and the editor, but as written the introduction still pitches this study as a use case for a technological method. I don't think this is the strongest argument and I am not sure the study as designed is really set up to provide a technical evaluation of the general capabilities of spatial transcriptomics. There are many studies deploying these methods and among these are a variety of studies specifically assessing their technical capabilities. In my view, the value of this study is in the biological insights it provides for understanding *Fasciola* and providing insight into potential drug or vaccine targets to further control of fascioliasis. This also provides an example of how these approaches can be used to further understanding of helminth biology. I am less compelled by the argument that this shows these methods can be used in non-model organisms. I don't know that there was any major question that these methods have broad use, and there are many studies that deploy spatial omics generally in non-model systems. From a technical perspective, this is another study using these methods. That isn't a criticism of the techniques, but rather illustrating my point that the value of the study, and its impact should be focused on the biological insights the data provides in the context of the parasite being studied and less on the technology. What are the major biological questions the authors are

seeking to understand and how have they succeeded in exploring or addressing these questions in their study? If there are specific technical hurdles that the authors overcame to be able to adapt these methods to this parasite, then that should be elaborated on. But as is, the focus on the technological advance is somewhat vague and doesn't really fit (in my opinion) with the focus of the manuscript, which is otherwise, really, looking that the tissue-specific differences in transcription the spatial omic approach provides and how this yields valuable insight into Fasciola and its control.

Lines 42-61 of the introduction touch on the biological relevance and focus of the study, but it's still somewhat generic and I find it difficult to really link these statements to what is explored in detail in the study. For example, if (based on the section from lines 41-52) the focus is on genes that are transcriptionally enriched in the tegument and intestine, as these are attractive as drug targets, why focus so much of the results section on reproductive tissues (e.g., lines 120-201)? To be clear, I think the more detailed tissue-specific results presented in the results is appropriate and I agree with their being included (indeed, I asked for this in my previous review). The issue I have is that the introduction doesn't give context to these results. The introduction is focused on the technology, but the results and discussion mostly focus on the biology. There is a disconnect here.

Lines 131-133: I'm not sure I understand what the authors are saying here. How does the spatial transcriptomic data they have generated show that development of *F. hepatica* oocytes also depend on 'biosynthetic activity and stored transcripts'? The authors need to provide a bit more context here.

Lines 164-192: I like the inclusion of the Giotto analysis in the study. But as written the current block of text feels out of context and is confusing. This overall section (lines 120-201) is supposed to be focused on the reproductive tissues (according to the section subheading), but expands into broader examination of the dataset as a whole, touches on some gene groups that are enriched in different tissues and then eventually moves back to reproductive tissues. It's a little confusing. I suggest present the global Seurat and Giotto analyses and how they compare / complement each other in allowing the authors to define tissue-specific transcriptomic profiles. Then decide which aspects of this (e.g. gonadal tissues etc) to explore in greater detail.

I think unfortunately the overall structure of the manuscript needs quite a bit of attention. I am pleased to see the authors have undertaken more detailed analysis of their data and I think these larger observations greatly improve the quality and impact of the study. However, as written the manuscript is quite hard to follow. As noted above, the introduction doesn't really provide the context the reader needs to understand the major focus of the study or to appreciate the impact of the major findings. The results section provides a deeper look at the dataset, which is definitely

valuable and should be included. But the sections are currently somewhat muddled and not in a particularly consistent order. For example the inclusion of the general description of the Giotta analyses in the middle of the description of the gonadal tissue section. The authors move from this to a focus on 'Vaccine candidates' in the gut, tegument and parenchyma (lines 202 onward). But, what about the drug targets alluded to in the introduction? And if that is the primary focus of this section, why provide a detailed presentation of the overall tissue transcriptomes, and not simply focus on the putative vaccine candidates? It's not that providing this information is inappropriate, it's that the order of presentation seems out of order or isn't given enough context.

My suggestion is that the authors revisit how they are presenting the study. I think the introduction needs to focus on the biological goal and primary questions (i.e., the importance of Fasciola and fascioliasis, the major gaps in our knowledge of its molecular biology and how this impedes new treatments against it). For there, the introduction can discuss the relevance of technical limitations of current approaches and explain how spatial omics can address this (i.e., providing tissue-specific transcriptomes in context etc). In my opinion, the results sections need to be condensed and restructured. First present the overall dataset and their suitability in resolving different transcriptomic clusters and the relationship these clusters have to the various major organs/tissues in the parasite. From there, present the advanced understanding of the transcriptomes of key organs and tissues (e.g., gonadal tissue, tegument, parenchyma and gut tissues). Then, explain the link between tissue-specific transcription profiles and suitability as drug/vaccine candidates and your findings on this. These elements are present, but they are somewhat scattered and the introduction does not provide the context to tie them together.

Overall, I still think the study is interesting and I think the additional analyses are interesting. Ultimately it is up to the editors and the authors as to whether the manuscript should be revised and restructured as I've suggested. This is of course just my opinion. But as is, I think the manuscript is improved significantly with the additional analyses, but hard to follow as written and the major impact of the study does not get the emphasis I think it needs. The biology is interesting and the data does provide major new insights that are compelling. The technology is interesting. However, although spatial transcriptomics is still a developing field and powerful technology, I don't think the current study stands on its technical advances alone.

Reviewer #3 (Remarks to the Author):

I would like to commend the authors on their thorough and considered response to all of the reviewers. All of my comments were addressed, and overall, I think the revised manuscript is greatly improved.

Well done. This will be a great resource for Fasciola research. I look forward to seeing it in print

.

Reviewer #3 (Remarks on code availability):

I could not access the code - the link was wrong.

Reply to reviewer comments (2nd revision)

for the manuscript “Spatial transcriptomics of a parasitic flatworm provides a molecular map of vaccine candidates, drug targets and drug resistance genes”

We thank the reviewers for their critical and helpful comments. We revised the introduction and the structure of the results section according to the reviewer suggestions. As requested, we reply point by point (text in blue). Line numbers in our reply refer to the line numbers of the “tracked changes” version of the manuscript.

Reviewer #1 (Remarks to the Author):

I have now read the authors' comments and appreciate the time they have taken to address my questions. The current version is a more robust manuscript and I hope the authors found my suggestions helpful. I am happy to support this manuscript for publication and I look forward to seeing this study in print.

Best wishes and good luck in your future studies!

We like to thank the reviewer for this positive feedback!

Reviewer #2 (Remarks to the Author):

Typo line 56 should be "drug and vaccine targets"

Revised as suggested (line 46)

Lines 67-71: Do the authors come back to these questions in their paper?

Our aim in the introduction was to raise major open questions regarding liver fluke biology. In the results, we deal with molecules that are potentially important for survival and relevant for host-parasite interaction in the context of drug and vaccine target prioritization. In the gonad chapter, we take a closer look at molecular mechanisms of reproduction. So yes, we come back to these questions, but of course we are far from being able to answer them completely. Therefore, we kept the questions in the introduction, but condensed the questions (lines 47ff.).

Lines 81-84: I think it would be better for this sentence to focus on tangible outcomes of the study.

Paragraph extended and concretized (lines 77ff.)

Introduction: it's ultimately up to the authors and the editor, but as written the introduction still pitches this study as a use case for a technological method. I don't think this is the strongest argument and I am not sure the study as designed is really set up to provide a technical evaluation of the general capabilities of spatial transcriptomics. There are many studies deploying these methods and among these are a variety of studies specifically assessing their technical capabilities. In my view, the value of this study is in the biological insights it provides for understanding *Fasciola* and providing insight into potential drug or vaccine targets to further control of fascioliasis. This also provides an example of how these approaches can be used to further understanding of helminth biology. I am less compelled by the argument that this shows these methods can be used in non-model organisms. I don't know that there was any major question that these methods have broad use, and there are

many studies deploy spatial omics generally in non-model systems. From a technical perspective, this is another study using these methods. That isn't a criticism of the techniques, but rather illustrating my point that the value of the study, and its impact should be focused on the biological insights the data provides in the context of the parasite being studied and less on the technology. What are the major biological questions the authors are seeking to understand and how have they succeeded in exploring or addressing these questions in their study? If there are specific technical hurdles that the authors overcame to be able to adapt these methods to this parasite, then that should be elaborated on. But as is, the focus on the technological advance is somewhat vague and doesn't really fit (in my opinion) with the focus of the manuscript, which is otherwise, really, looking at the tissue-specific differences in transcription the spatial omic approach provides and how this yields valuable insight into Fasciola and its control.

Lines 42-61: of the introduction touch on the biological relevance and focus of the study, but it's still somewhat generic and I find it difficult to really link these statements to what is explored in detail in the study. For example, if (based on the section from lines 41-52) the focus is on genes that are transcriptionally enriched in the tegument and intestine, as these are attractive as drug targets, why focus so much of the results section on reproductive tissues (e.g., lines 120-201)? To be clear, I think the more detailed tissue-specific results presented in the results is appropriate and I agree with their being included (indeed, I asked for this in my previous review). The issue I have is that the introduction doesn't give context to these results. The introduction is focused on the technology, but the results and discussion mostly focus on the biology. There is a disconnect here.

We thank the reviewer for this detailed feedback on our introduction. While the method is innovative and new for the field of helminthology, which is what we wanted to highlight, we indeed recognize the reviewer's point. We have now revised the introduction and tried to incorporate the suggestion to focus more on liver fluke biology and drug and vaccine discovery rather than the technology.

Lines 131-133: I'm not sure I understand what the authors are saying here. How does the spatial transcriptomic data they have generated show that development of *F. hepatica* oocytes also depend on 'biosynthetic activity and stored transcripts'? The authors need to provide a bit more context here.

It was more an assumption based on our finding that the ovary has high levels of transcripts and GO terms are enriched for biosynthetic processes. Of course, we cannot confirm in how far the oocytes and the development of the embryo are actually "dependent" on these processes, as this was also not the purpose of our study. The statement was removed.

Lines 164-192: I like the inclusion of the Giotto analysis in the study. But as written the current block of text feels out of context and is confusing. This overall section (lines 120-201) is supposed to be focused on the reproductive tissues (according to the section subheading), but expands into broader examination of the dataset as a whole, touches on some gene groups that are enriched in different tissues and then eventually moves back to reproductive tissues. It's a little confusing. I suggest present the global Seurat and Giotto analyses and how they compare / complement each other in allowing the authors to define tissue-specific transcriptomic profiles. Then decide which aspects of this (e.g. gonadal tissues etc) to explore in greater detail.

Thanks for this suggestion to revise and restructure the results part. We now introduce the Giotto analysis right after the Seurat chapter and then go into more detail. We realize that this is the more logical structure to follow (lines 142 ff).

I think unfortunately the overall structure of the manuscript needs quite a bit of attention. I am pleased to see the authors have undertaken more detailed analysis of their data and I think these larger observations greatly improve the quality and impact of the study. However, as written the manuscript is quite hard to follow. As noted above, the introduction doesn't really provide the context the reader needs to understand the major focus of the study or to appreciate the impact of the major findings. The results section provides a deeper look at the dataset, which is definitely valuable and should be included. But the sections are currently somewhat muddled and not in a particularly consistent order. For example the inclusion of the general description of the Giotta analyses in the middle of the description of the gonadal tissue section. The authors move from this to a focus on 'Vaccine candidates' in the gut, tegument and parenchyma (lines 202 onward). But, what about the drug targets alluded to in the introduction? And if that is the primary focus of this section, why provide a detailed presentation of the overall tissue transcriptomes, and not simply focus on the putative vaccine candidates? It's not that providing this information is inappropriate, it's that the order of presentation seems out of order or isn't given enough context.

My suggestion is that the authors revisit how they are presenting the study. I think the introduction needs to focus on the biological goal and primary questions (i.e., the importance of Fasciola and fascioliasis, the major gaps in our knowledge of its molecular biology and how this impedes new treatments against it). For there, the introduction can discuss the relevance of technical limitations of current approaches and explain how spatial omics can address this (i.e., providing tissue-specific transcriptomes in context etc). In my opinion, the results sections need to be condensed and restructured. First present the overall dataset and their suitability in resolving different transcriptomic clusters and the relationship these clusters have to the various major organs/tissues in the parasite. From there, present the advanced understanding of the transcriptomes of key organs and tissues (e.g., gonadal tissue, tegument, parenchyma and gut tissues). Then, explain the link between tissue-specific transcription profiles and suitability as drug/vaccine candidates and your findings on this. These elements are present, but they are somewhat scattered and the introduction does not provide the context to tie them together.

Many thanks for these ideas to further improve our manuscript. In the first submission, we had clearly separated the presentation of individual tissue transcriptomes from the presentation of gene families important for applied research. We believed to understand from a previous reviewer comment that it would be preferred to create better connections between both parts. This resulting "muddle" was not ideal, as you pointed out. We are therefore happy to restructure the results chapter once more with clear separation and think the data presentation is now the best possible solution.

Overall, I still think the study is interesting and I think the additional analyses are interesting. Ultimately it is up to the editors and the authors as to whether the manuscript should be revised and restructured as I've suggested. This is of course just my opinion. But as is, I think the manuscript is improved significantly with the additional analyses, but hard to follow as written and the major impact of the study does not get the emphasis I think it needs. The biology is interesting and the data does provide major new insights that are compelling. The technology is interesting. However, although spatial transcriptomics is still a developing field and powerful technology, I don't think the current study stands on its technical advances alone.

We greatly appreciate the reviewer's valuable suggestions and constructive criticism, this is well-received!

Reviewer #3 (Remarks to the Author):

I would like to commend the authors on their thorough and considered response to all of the reviewers. All of my comments were addressed, and overall, I think the revised manuscript is greatly improved.

Well done. This will be a great resource for Fasciola research. I look forward to seeing it in print.

We thank the reviewer for this positive feedback!

Reviewer #3 (Remarks on code availability):

I could not access the code - the link was wrong.

We apologize for this inconvenience. The DOI given in the manuscript to access Zenodo is not yet active and will only be activated after publication. We have created an additional DOI for reviewer access (10.5281/zenodo.10302804). We apologize if this DOI was not available for reviewers.

REVIEWERS' COMMENTS

Reviewer #2 (Remarks to the Author):

Yes! Thank you to the authors for their further revision and restructuring of the manuscript. This looks much much better in my opinion. It presents the study narrative much more clearly, makes the findings far more accessible and adds the needed biological context that I think was not strong enough in the earlier version. Well done! I'm really pleased with the revised manuscript, which has now addressed all of my major comments. Excellent work.

Three very minor comments below that the authors can consider, but do not need to be reviewed again by me.

line 176-177: Something missing in this sentence. Please check.

Various locations throughout manuscript: my preference would be for the authors to use the terms 'transcription/transcribed' instead of 'expression/expressed'. Expression infers protein production. In the present manuscript, I think this distinction is particularly important. The spatial data allows the authors to explore the tissue-specific transcriptional differences within *Fasciola*. Although it is reasonable to expect the expression patterns of most proteins will reflect their tissue-specific transcription, this is not always the case. I am not sure how much this has been explored in flukes, but in nematodes, for example, there are a variety of genes that are transcribed in gut tissues, but transported elsewhere as proteins. In addition, transcripts may be stabilised and translational repressed in tissues by RNA binding proteins. This is a key mechanism through which eukaryotic cells program cell development, including in stem cells etc. Thus, detection of a transcript does not always infer the present of the protein it encodes (at least not contemporarily). This is not a deficiency of the study, and the authors need not go into this in their manuscript, but I think it would be best to leave open the possibility that some *Fasciola* genes may be transcribed in one tissue and transported elsewhere for expression, or transcribed but not expressed until later on. However, expression is often used as synonymous with transcription, so if the authors prefer to leave the manuscript as is, it isn't incorrect as written.

Line 417-418: Noting my comment above, please clarify whether the proteins were localised or their transcripts.

Reply to reviewer comments (3rd revision)

for the manuscript “Spatial transcriptomics of a parasitic flatworm provides a molecular map of vaccine candidates, drug targets and drug resistance genes”

We thank the reviewer for their critical and helpful comments. We addressed the remaining points according to the reviewer suggestions. As requested, we reply point by point (text in blue). Line numbers in our reply refer to the line numbers of the “tracked changes” version of the manuscript.

Reviewer #2 (Remarks to the Author):

Yes! Thank you to the authors for their further revision and restructuring of the manuscript. This looks much much better in my opinion. It presents the study narrative much more clearly, makes the findings far more accessible and adds the needed biological context that I think was not strong enough in the earlier version. Well done! I'm really pleased with the revised manuscript, which has now addressed all of my major comments. Excellent work.

Three very minor comments below that the authors can consider, but do not need to be reviewed again by me.

line 176-177: Something missing in this sentence. Please check.

- This point has indeed been overlooked when preparing the 2nd revision. Gaps were now filled with the missing numbers of genes (line 149f.): “Of the 79 genes in the list, 55 genes were expressed above average in the testis and even 74 in the ovary.”

Various locations throughout manuscript: my preference would be for the authors to use the terms 'transcription/transcribed' instead of 'expression/expressed'. Expression infers protein production. In the present manuscript, I think this distinction is particularly important. The spatial data allows the authors to explore the tissue-specific transcriptional differences within *Fasciola*. Although it is reasonable to expect the expression patterns of most proteins will reflect their tissue-specific transcription, this is not always the case. I am not sure how much this has been explored in flukes, but in nematodes, for example, there are a variety of genes that are transcribed in gut tissues, but transported elsewhere as proteins. In addition, transcripts may be stabilised and translational repressed in tissues by RNA binding proteins. This is a key mechanism through which eukaryotic cells program cell development, including in stem cells etc. Thus, detection of a transcript does not always infer the present of the protein it encodes (at least not contemporarily). This is not a deficiency of the study, and the authors need not go into this in their manuscript, but I think it would be best to leave open the possibility that some *Fasciola* genes may be transcribed in one tissue and transported elsewhere for expression, or transcribed but not expressed until later on. However, expression is often used as synonymous with transcription, so if the authors prefer to leave the manuscript as is, it isn't incorrect as written.

Line 417-418: Noting my comment above, please clarify whether the proteins were localised or their transcripts.

- As this is a transcriptomics study, the term “expression” is used synonymous with transcription in this manuscript. A statement clarifying this aspect was now added at the beginning of the results part (line 71f.): “In this manuscript, the term “expression” refers to transcript levels, not protein levels.”